# Pan-cancer classification of single cells in the tumour microenvironment

Ido Nofech-Mozes [1,2], David Soave[1,3], Philip Awadalla [1,2,4,5] ✉ & Sagi Abelson [1,2,5] ✉

Single-cell RNA sequencing can reveal valuable insights into cellular heterogeneity within tumour microenvironments (TMEs), paving the way for a deep understanding of cellular mechanisms contributing to cancer. However, high heterogeneity among the same cancer types and low transcriptomic variation in immune cell subsets present challenges for accurate, high-resolution confirmation of cells' identities. Here we present scATOMIC; a modular annotation tool for malignant and non-malignant cells. We trained scATOMIC on >300,000 cancer, immune, and stromal cells defining a pan-cancer reference across 19 common cancers and employ a hierarchical approach, outperforming current classification methods. We extensively confirm scATOMIC's accuracy on 225 tumour biopsies encompassing >350,000 cancer and a variety of TME cells. Lastly, we demonstrate scATOMIC's practical significance to accurately subset breast cancers into clinically relevant subtypes and predict tumours' primary origin across metastatic cancers. Our approach represents a broadly applicable strategy to analyse multicellular cancer TMEs.

Tumour microenvironments (TMEs) are highly complex. Various immune and stromal cells within the TME interact with cancer cells to regulate processes such as angiogenesis, tumour proliferation, invasion, and metastasis, as well as mediate mechanisms of therapeutic resistance[1–4]. Single-cell RNA sequencing (scRNA-seq) techniques are explicitly suitable to disentangle complex systems as they provide transcriptome information for every cell within a sample, enabling the study of subtle transcriptomic changes reflecting different cell types and their functional states[5].

Cell-type annotation is arguably the most critical step to derive biological insight from scRNA-seq experiments and can be performed manually or using automatic classifiers[6,7]. Manual annotation is often unfavourable as it is subjective to user definition of non-parametric clustering of cells, conducted under the assumption that all the cells within a defined cluster are identical, and depends on pre-existing knowledge of canonical genes. Although expression of canonical markers has been used to characterise some cell types, definitive markers are not always available[8]. Moreover, due to their

relatively low number and the possibility of incomplete detection due to technical variation, the sole use of canonical gene expression is not ideal.

Given these limitations, there has been a shift towards automatic methods for cell classification, with over 100 classifiers described in a recent census of available scRNA tools[9]. To date, most automated annotators are focused on classification of blood or subsets of cells from other specialised tissues, thus having limited capabilities in deciphering complex TMEs across diverse human cancers. Indeed, using single-cell transcriptomics to predict cancer types and differentiate between cancer and related normal tissue cells while also classifying the large number of immune cells and stroma, is not a straightforward task[10]. In the context of TMEs, cell type predictions are challenged by high inter-patient tumour cell heterogeneity among cancers of the same tissue[11–13] and low transcriptomic variation among related, yet different specialised immune cells[14]. Since cancer samples tend to cluster by patient[11–13] and transcriptional variation is often driven by genomic instability, existing cell type classification methods

[1]Ontario Institute for Cancer Research, Toronto, ON, Canada. [2]Department of Molecular Genetics, University of Toronto, Toronto, ON, Canada. [3]Department of Mathematics, Wilfrid Laurier University, Waterloo, ON, Canada. [4]Dalla Lana School of Public Health, University of Toronto, Toronto, ON, Canada. [5]These authors jointly supervised this work: Philip Awadalla, Sagi Abelson. ✉e-mail: philip.awadalla@oicr.on.ca; sabelson@oicr.on.ca

which rely on distance correlations to a reference[15–17] are expected to fail.

The current standard for the identification of malignant cells in scRNA-seq data relies on copy number variation (CNV) inference methods[18,19]. Nevertheless, these methods are incapable of providing definitive information concerning the cancer's tissue of origin. Furthermore, CNV inference necessitates the presence of genetically unstable cells, and its accuracy may suffer when lacking a sizeable, distinctive normal cell reference within the sequenced specimen. Solely relying on the presence of inferred CNVs to annotate malignant cells may lead to false negatives or undefined cells in cancers with minimal genomic structural variation or nearly diploid genomes. Thus, a limitation in scRNA-seq analysis of tumour ecosystems is that there is no universal method for effective, detailed classification of heterogenous non-malignant TME cell types and subtypes, and cancer cells.

Clearly, a fully automated, pan cancer classification scheme that can easily be updated to capture additional subsets of normal cells and clinically relevant molecular subtypes of cancer, holds promise towards a better understanding of cancer ontogenies and the molecular interaction of diverse tumour tissues with their microenvironments.

In this work, we present **s**ingle **c**ell **a**nnotation of **t**um**o**ur **m**icro-environments **i**n pan-**c**ancer settings (scATOMIC), a comprehensive, pan cancer, TME cell type classifier. We devise a structured scheme that uses hierarchically organized models and elimination processes, reducing the transcriptomic complexity of the TME multi-cellular system to improve cell classification.

## Results

### An overview of scATOMIC

We postulated that the sheer number of publicly available single-cell transcriptomic datasets will enable the development of a highly accurate and thorough classifier for cancer, blood, and stromal cells. To define a pan cancer reference, we interrogated cancer patient data, augmented by two additional comprehensive data sources containing transcriptomic-independent confirmation of cell identities. These include scRNA-seq of cancer cell lines representing 19 common cancer types[20] and a CITE-seq dataset (proteomics and transcriptomics) of diverse peripheral blood cells[16]. scRNA-seq of stromal cells was gathered from several tumour and normal tissue sources[21–28] (Supplementary Data 1, 2). Overall, 301,662 cells were included in the training reference dataset of scATOMIC.

Obtaining an accurate set of differentiating features is critical to successful classification. Nonetheless, significant differentially expressed genes (DEG) concerning non-malignant cell types are often expressed in other related cells that are functionally distinct[14,29,30] (Supplementary Fig. 1). On the other hand, inter-patient heterogeneity among malignant cells has been repeatedly observed with different patients forming unique clusters[11–13] (Supplementary Fig. 2). To improve cell identity predictions, we developed a method, termed reversed hierarchical classification and repetitive elimination of parental nodes (RHC-REP) which reduces the breadth of cell types in an ensemble of classification tasks. As compared with top-down local hierarchical methods, here, predictions of terminal classes are repeatedly being evaluated to infer the cells' broader parental nodes. During this process terminal cell classes are investigated iteratively using multiple sets of refined differentiating features until confident terminal annotations are achieved.

To develop this approach, we structured a pan cancer TME cellular hierarchy where each parent node represents a group of related cells, and each terminal node represents a single-cell class of interest. Overall, we trained 24 random forest models corresponding to the total number of parent nodes (Fig. 1a). For every model, we selected DEGs that distinguish each cell type from all other terminal classes nested within the same parent. RHC-REP will then prioritise the

features with the highest specificity to the interrogated cell types (Fig. 1b).

During each classification task, every cell receives a vector of prediction scores (PS) corresponding to the percentage of trees voting for each terminal class in the parent node (Fig. 1c). This cell by PS matrix is then used to calculate intermediate group scores (IGS), to subsequently link cells to their next parental node in the hierarchy (Fig. 1d, Supplementary Fig. 3). At each classification task, the distribution of IGSs obtained from all the cells interrogated in the model is used to automatically define prediction cut-offs (Supplementary Fig. 4). Each cell is then interrogated by its next associated model, defined by a more discriminative set of features and fewer potential terminal classes (Fig. 1e). Cells that do not pass the IGS thresholds are given their previous parent classification and withheld from further subclassification (Supplementary Note 1).

Given that non-malignant cells that share the cancer's tissue of origin can be found in cancer biospecimens (for example, normal alveolar cells in a lung biopsy) we embedded a cancer signature scoring and cell differentiating module in scATOMIC. Using an established transcriptional program scoring method[11,16], cancer-type-specific up and down-regulated programs[31] are evaluated in cells receiving an original annotation of a cancer type by scATOMIC (Fig. 1f).

### Performance evaluation and validation across internal and external datasets

To evaluate scATOMIC's performance, we first conducted fivefold cross validation using the training reference dataset (Supplementary Data 1) while keeping equal proportions of cell types in each of the five folds. scATOMIC achieved median F1 scores ranging from 0.90 to 0.99 across all the tested cell types (Fig. 2a), implying great accuracy in classifying the breadth of cells in the settings of pan cancer TMEs. To ensure scATOMIC was not heavily impacted by batch effects, we also trained 4 scATOMIC iterations, where all cells from intact technical batches were held out from training. We found no significant difference in the performance of the models on the held out batches (Supplementary Fig. 5a, Kruskal–Wallis rank sum test: H statistic(degrees of freedom = 3) = 6.12, P value = 0.106). We further tested scATOMIC performance across different scRNA platforms using external melanoma datasets[29,32–34] and again, found no significant difference in F1 scores (Supplementary Fig. 5b, Kruskal–Wallis rank sum test: H statistic(degrees of freedom = 3) = 2.18, P value = 0.537).

We next aimed to conduct a comprehensive external, training-independent validation of scATOMIC performance. To build a validation dataset with high-confidence cell annotations, we mined publicly available scRNA-seq data from primary tumour biopsies and blood samples. Overall, the curated set used for validation contained 228,460 cancer, 82,976 stroma and 46,090 blood cells from 225 primary biopsies spanning 13 cancer types (Supplementary Data 3). Importantly, these ground truth sets include cancer cells supported by abnormal CNV profiles, and immune cells with transcriptomic-independent identity supported by cell surface protein markers via CITE-seq. Similar to the results obtained from internal validation, in this independent validation process, scATOMIC achieved a median F1 score of 0.99 (Fig. 2b).

Overall, these results demonstrate the broad abilities of scATOMIC's core algorithm to detect cancer cells and their type, as well as predicting non-malignant cell types and subtype.

### Comparison with other cell-type classification methods

We compared scATOMIC's performance to six commonly used scRNA-seq classifiers (SingleR[15], Seurat[16,35], SingleCellNet[36], scmap-cell[17], CHETAH[37], and scType[38]). These annotators encompass a variety of methods including reference correlation, label transfer using integration to a reference, random forest algorithm, flat and hierarchical models, and marker-based classification methods making their

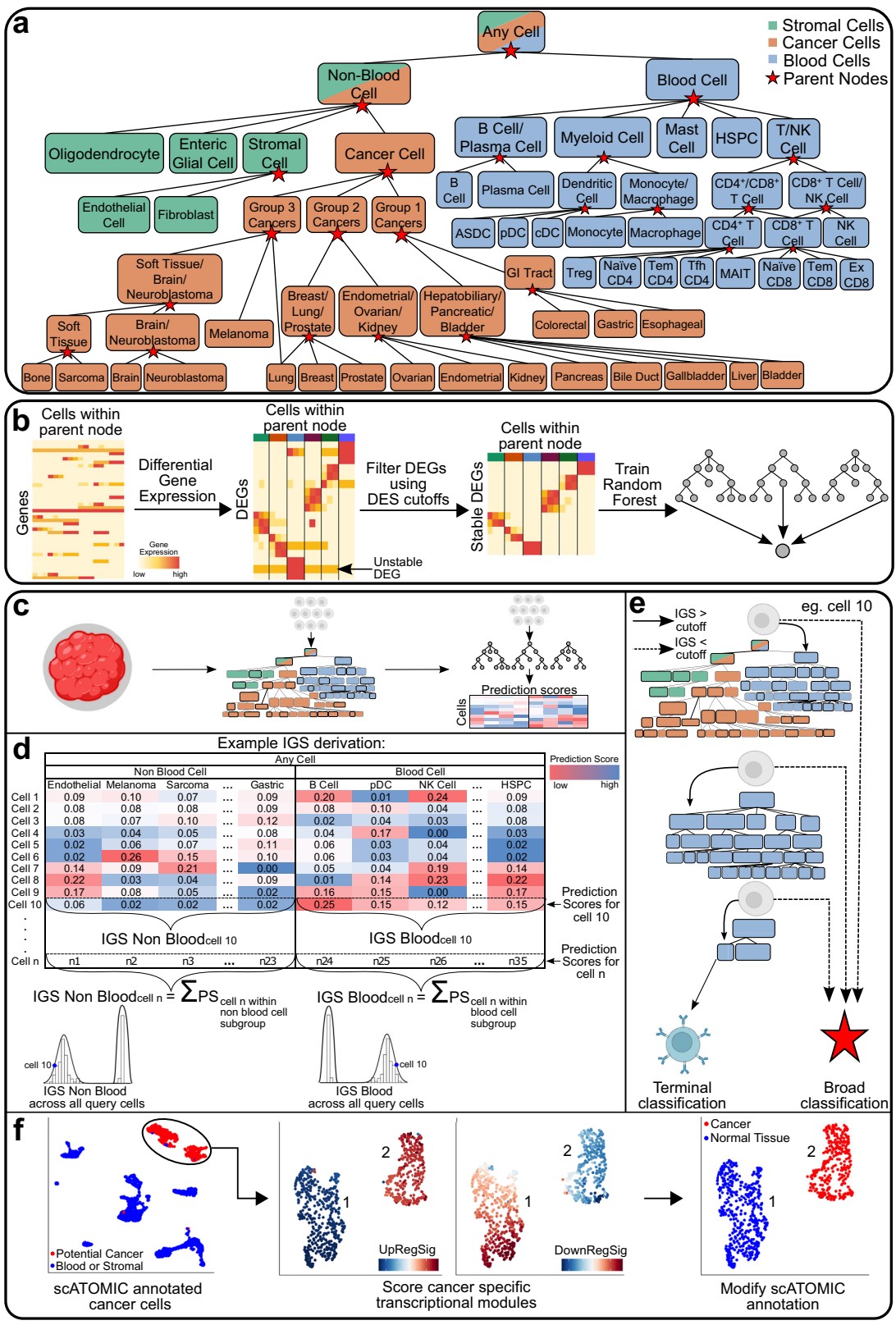

comparison with scATOMIC's underlying RHC-REP approach highly suitable. Each tool was provided with the same reference and external validation dataset as scATOMIC for training and testing (Supplementary Data 1–3). All classifiers were highly accurate in classifying non-malignant cells (median F1 > 0.85, Fig. 2c). However, for cancer cells in particular, F1-scores were significantly lower as compared with scATOMIC (two-sided Wilcoxon rank sum tests, all *P* values < 2 × 10⁻¹⁶). The

next best performing classifier following scATOMIC was SingleR[15] with median F1 scores of 0.92, 0.97 and 0.76, for blood, stromal, and cancer cells, respectively (Fig. 2c). Using scATOMIC, we obtained median F1 scores of 0.95 0.99, and 0.99 in each of these respective categories. As existing cell type classification tools were not designed to annotate malignant cells, this comparison highlights scATOMIC's ability to overcome the complexity presented in pan cancer settings to

**Fig. 1 | Overview of scATOMIC training and classification. a** Hierarchical structure of the pan-cancer tumour microenvironment. The cellular hierarchies in the pan-cancer tumour microenvironment are organized into a flow chart with increasing cell type resolution. Parent nodes represent broad classification branches, and terminal nodes represent specialised cell classes of interest. **b** Training of classification branches for each parent node ($n = 24$). The reference datasets are filtered based on transcriptomic-independent information to only include terminal cell types that are found within a particular parental node. Genes that significantly differentiate one cell type from all the others are gathered. Differentially expressed genes (DEGs) with greater specificity to each terminal class, determined by differential expression score (DES), are kept (Methods). A random forest classifier is trained on filtered, library size normalised count matrices to derive a model that provides prediction scores corresponding to the proportion of trees voting for each terminal class within the parental node. Colours on the top of the heatmap illustrate different cell types. **c–f** Classification of query datasets. **c** Gene expression count matrices from query tumour biopsies are inputted into the first scATOMIC classification branch model, outputting a cell-by-prediction scores matrix. **d** Prediction scores (PS) from all blood and non-blood cell subtypes are respectively summed to derive intermediate group score (IGS) distributions associating single cells with their appropriate parental class. **e** Cells are iteratively interrogated at their next parent nodes' corresponding models until terminal classification are obtained. Broad classifications occur if the IGS for a cell is lower than the confidence cut-off. In this example, cell 10 is subclassified until a terminal B cell designation is derived. **f** Differentiating between cancer and tissue-specific non-malignant cells through scoring of bulk RNA-seq derived differentiating gene expression programs (Methods). scATOMIC automatically annotates population 2 as cancer cells, and population 1 as non-malignant. Heatmaps and cell illustrations were created with BioRender.com.

accurately identify cancer cells while also having comparable or significantly better performance in annotating stroma and blood. Lastly, with the exception of CHETAH[37], mostly comparable time and memory usages were measured for all the methods (Supplementary Fig. 6).

## Distinguishing between non-malignant, tissue-specific cells and cancer

Aneuploid CNV profiles are highly associated with the development and progression of numerous cancers by impacting gene expression levels[39]. We assessed scATOMIC's ability to distinguish malignant cells from other normal cells of the TME, sharing the same cell-of-origin, by comparing scATOMIC's final cancer predictions (Fig. 1f) to their inferred CNV-based ploidy status[19]. We observed strong agreement in cells predicted as malignant and aneuploid-inferred CNV profiles across biopsies, as well as between non-malignant detected cells and diploid inferred profiles, with a median agreement rate of 85.9% (Fig. 3). In addition, in silico serial dilution analysis of cancer cells in our external dataset collection revealed that decreasing number of malignant cells can be appropriately annotated, with scATOMIC also providing cancer type notations (Supplementary Fig. 7). Discordant cases, defined as cases with an agreement rate below the 1st quartile (Q1 ≤ 63.2%), were typically attributed to tumours with low CNV levels, intra-tumoural malignant subclones, low number of cancer cells harbouring the CNV, or low number of reference cells (Supplementary Fig. 8). In only 7% of discordant cases more cells were inferred as aneuploid than cells annotated as malignant by scATOMIC (Fig. 3b). These results suggest that cancer and related normal tissue cells are efficiently classified by their transcriptomic profiles using scRNA-seq data, independent of their ploidy status.

## scATOMIC annotations increase cellular resolution in tumour biopsies

To further demonstrate the benefits of scATOMIC in annotating multicellular TMEs, we analysed several datasets, including scRNA-seq of lung cancer[29]. Original annotations for this dataset were determined by the authors using SingleR[15] with its default references in combination with cell type signatures and the use of canonical marker genes. Similar to our observations (Supplementary Fig. 1) Slyper et al.[29], noted overlapping expression programs between T cells and NK cells which makes high resolution single-cell discrimination among these cell types challenging. scATOMIC resolved NK cells and T cells, and further subclassified the latter into fine grained subtypes including T regulatory cells, naive CD4 + T cells, CD4 + T follicular helper cells, effector/memory CD4+, effector/memory CD8 + T cells, and exhausted CD8 + T cells (Fig. 4a). In addition, in this lung dataset, scATOMIC identified other distinct cell types including plasma cells, and plasmacytoid dendritic cells (pDCs) which scATOMIC separated from B cells. Unsupervised clustering and expression of cell specific markers[34] supported the existence of these separated cell identities (Supplementary Fig. 9). Of note, this biopsy included two more small clusters

of "epithelial cells" (Supplementary Fig. 9a), suggesting tissue-specific cell classes that are not represented in the current scATOMIC training reference. Using scATOMIC's automatic approach to set confidence IGS cut-offs (Supplementary Fig. 4) these cells were abstained from being falsely annotated and correctly assigned with the lower-level annotation of non-blood cells. scATOMIC resolved the remaining epithelial cells into lung cancer and normal tissue cells by evaluating lung cancer associated transcriptional signatures (Fig. 1f).

Increased cellular resolution across the cell types of the TME was also observed in other recent datasets of different cancer types including bladder[4], breast[40], liver[41], ovarian[40], prostate[42], and skin cancer[33] (Fig. 4b–g). In addition, scATOMIC identified hematopoietic stem/progenitor cells (HSPCs) in glioblastoma[43] (Supplementary Fig. 10); a population which was shown to promote tumour cell proliferation[43].

Collectively, this analysis demonstrates the ability of scATOMIC's core hierarchical algorithm to resolve cell identities at high resolution, label fine grained T cell states, identify rare cell types, abstain from falsely classifying unknown cells, and determine cancer types.

## Extending the core scATOMIC hierarchy for novel applications

By leveraging RHC-REP, one can easily deploy new scRNA-seq data to train extensions at any terminal branch of the hierarchy. We thought that extending the breast cancer classification node would provide a practical example of utilising modularity (Fig. 5a). Two sizable scRNA-seq breast cancer atlases were used to train, and independently test (Supplementary Data 4, 5) a classification model that resolves breast cancer cells into the major ER + , HER2 + and triple negative breast cancer (TN) histological subtypes. We applied scATOMIC to the training-independent validation dataset containing 38 tumours spanning ER + , HER2 + , and TN breast cancer, and 2 HER2 + /ER + double positive tumours, a class not represented in the current reference of scATOMIC's breast mode due to a lack of data. scATOMIC correctly subtyped 37 of the 38 (97.4%) training-independent breast cancer biopsies, as determined by immune-staining[21,44] (Fig. 5b, Supplementary Data 5). In the two HER2 + /ER + double positive samples, scATOMIC assigned mixed annotations of HER2 + and ER + cells (Fig. 5b).

We observed different degrees of tumour cellularity, with 6 biopsies (15%) having more predicted normal breast cells than cancer cells. In another tumour reported as ER[low] (that is, <10% ER + cancer cells by immunostaining), scATOMIC identified 8% ER + breast cancer cells (Fig. 5c, Supplementary Data 5). Of note, scATOMIC identified these ER + cells as malignant, in line with the histology report, however CNV inference predicted a diploid profile (Fig. 5d). This example highlights a distinct subpopulation of cancer cells that could have been misinterpreted as normal tissue by strictly relying on CNV inference, thus suggesting integrative approach for best results.

Overall, these data demonstrate scATOMIC's practical and modular framework to further subset primary tumour classes into their clinically relevant subtypes.

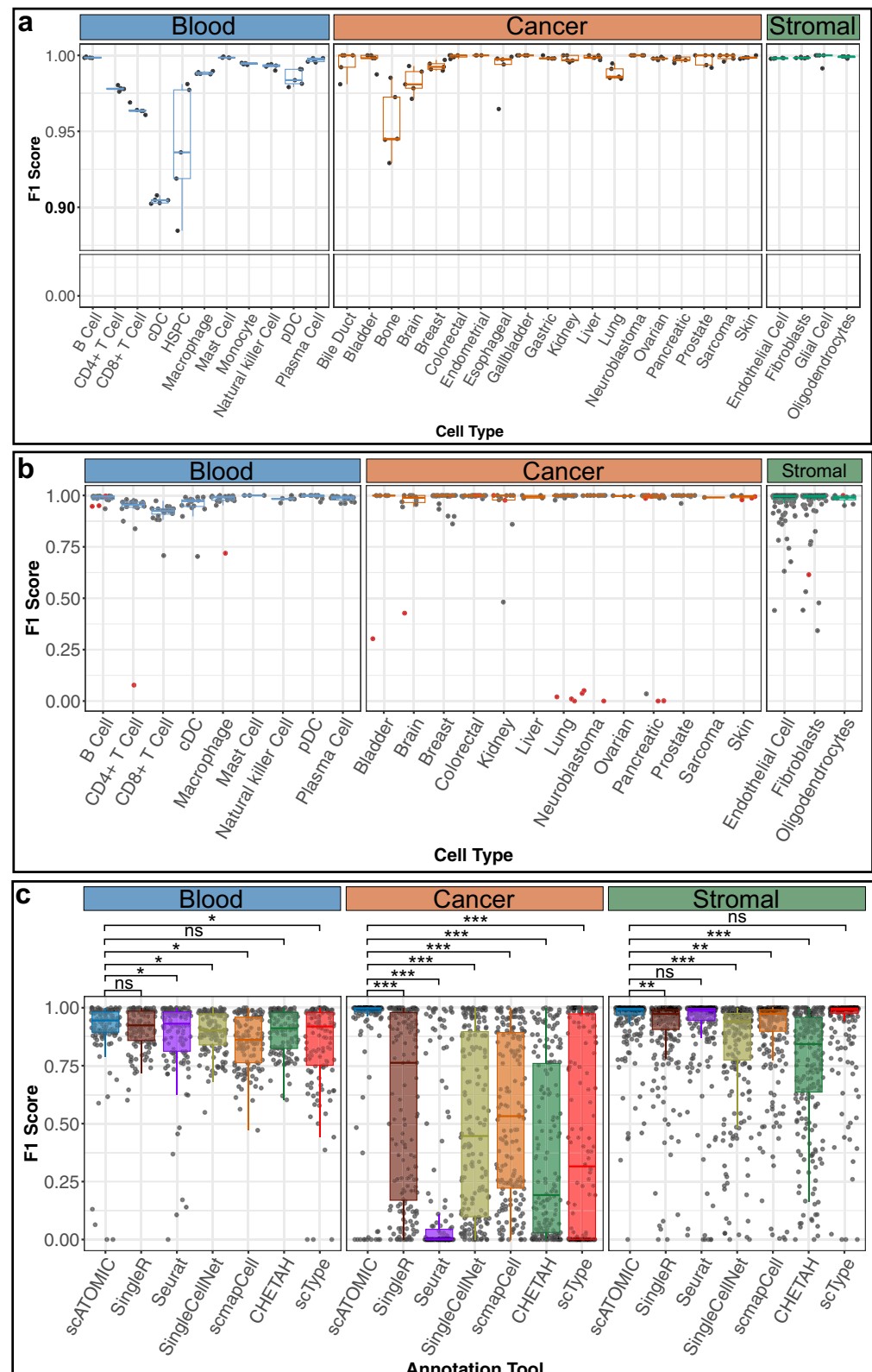

## scATOMIC identifies the tumour of origin across metastatic cancers

Given that existing single-cell annotation tools are not designed to provide information regarding the originating tissue of a cancer cell, we applied scATOMIC to predict the tumour origin in settings where it may be unknown. We curated a dataset of 62 metastatic biopsies from breast, kidney, lung, ovarian, and skin cancers from diverse anatomical

sites (Supplementary Data 6). In 52 of the 62 samples (83.9%), the primary tissue of origin was correctly predicted by scATOMIC (Fig. 6), demonstrating its robustness at distant sites, in cells that may have undergone transcriptional changes associated with metastasis. In 1 kidney and 2 lung samples (additional 4.9%) scATOMIC abstained from giving a terminal classification yet focused the prediction on the correct intermediate class. In 2 lower throughput melanoma scRNA-seq,

**Fig. 2 | scATOMIC performs accurately in internal and external validation experiments. a** *k*-fold cross validation. The reference dataset was randomly split into 5 sub-samples containing equal numbers of each cell type. F1 scores are shown for each cell type in each of the 5 replicates (jitter points). Each fold contained overall ~61,100 cells. Boxplot colours represent the major cell type classes. **b** External validation in datasets not used for training. scATOMIC was validated on CITE-seq datasets of tumour derived blood cells, datasets of aneuploid cancer cells and stromal cells from primary tumour biopsies. F1 scores are shown for each cell type within individual samples (jitter points). The plot represents *n* = 357,526 cells from 225 samples. Red dots indicate low-confidence cell type classifications (Methods). Boxplot colours represent the major cell type classes. **c** scATOMIC

outperforms other existing automatic cell type annotators, particularly when applied to identify cancer cells and determine their type. Six existing classifiers were provided the same training/reference and training-independent validation datasets as scATOMIC. Combined F1 scores for each of the three major cell class, blood, cancer, and stroma are shown (jitter points). The plot represents *n* = 337,790 cells from 221 samples that were given a classification output by all tools. (two-sided Wilcoxon rank sum test comparing scATOMIC to each tool *$P < 0.05$, **$P < 1.1 \times 10^{-6}$, ***$P < 2 \times 10^{-16}$, are shown). Boxplot colours represent the different tools. For all plots, boxes and whiskers represent the lower fence, first quartile (Q1), median (Q2), third quartile (Q3), and upper fence. Source data are provided as a Source Data file.

only 5 and 6 cancer cells were reported[11] yet scATOMIC found none. We considered these as false predictions. In 4 of the 5 remaining samples that received incorrect terminal classifications, the predicted cancer type and the reported primary were related cancers falling under the same immediate parent node. For example, a mixed serous/clear-cell ovarian carcinoma was predicted to be endometrial cancer, with relatively low classification scores (Supplementary Fig. 11). Overall, these results show that accurate detection of metastatic cancers' tissue of origin using single-cell transcriptomics is feasible and that scATOMIC can aid in identifying cancers' primary sites across a variety of solid human tumours.

## Discussion

The rate of scRNA-seq publications reporting major scientific insights concerning the function of various immune and stromal cells in cancer has increased steadily over the years[40,45,46]. However, the lack of automated methods that can also standardise the identification of single malignant cells is becoming a major obstacle to accurately study tumour-microenvironment interactions across various cancer types.

We developed scATOMIC to effectively annotate the TME in pan-cancer settings. scATOMIC overcomes several classification challenges, including high inter-patient heterogeneity and highly overlapping expression profiling among specificized immune cells. By using stably expressed transcripts as features, structured classification, and models trained using reliable and large datasets, scATOMIC has proven to accurately identify cancer cells and their origin. Moreover, scATOMIC is comparable to or outperforms other existing automatic cell type annotators when classifying blood and stromal cells using our training reference. In samples with genome instability and an appropriate reference of normal cells, we found high agreement between scATOMIC and CNV inference to pinpoint malignant cells in scRNA-seq data. However, in samples with cancer sub-clones defined by variable CNV burdens, cancer cells with near diploid genomic profiles, or few normal cells to serve as controls, CNV inference may fall short. Since information concerning CNV may be useful for cancer prognostication, and a degree of discordancy still exists, using scATOMIC in conjunction with CNV inference to annotate cancer cells and their type is recommended.

We designed the core, ploidy-neutral scATOMIC algorithm to accurately identify cancer and normal tissue cells across 19 common cancer types, including key rare populations such as plasmacytoid dendritic cell and hematopoietic stem and progenitor cells that were reported in cancer tissues and are associated with immunosuppressive phenotypes[29,43]. To ensure that scATOMIC remains powerful, we designed it in a way that new data can be easily interrogated to extend the core hierarchy by adding new terminal cell classes. We demonstrated this modularity by further classifying breast cancers into their clinically relevant molecular subtypes achieving high agreement between transcriptomics and immunostaining. With the progressive accumulation of high quality publicly available scRNA-seq data, future extensions of the core hierarchy to further subclassify the other 18 cancer types and the various core, non-malignant cell types to their more resolved classes or states will become simple.

As molecular classification of cancers by tissue-of-origin is fundamental to diagnostic pathology we demonstrated scATOMIC's ability and high accuracy in predicting the primary origin of metastatic tumours. Additional work is required to evaluate the limits of single-cell transcriptomics to predict cancer origin, specifically in cancers of unknown primary and other contexts where distinguishing primary from metastatic tumours is not trivial, such as in the case of mucinous ovarian carcinoma[47].

In summary, we have described, benchmarked, and validated a highly accurate single-cell annotation tool across TMEs of common, deadly cancer types. The RHC-REP classification approach underlying scATOMIC used here to tackle the complexity associated with multi-cellular TMEs might be of interest in areas outside the cancer field entailing multi-class structured systems. We highlight the benefits that scATOMIC holds in cancer settings compared to other tools, providing a method to standardise single-cell cancer transcriptomic studies. We expect that scATOMIC's abilities to accurately identify TME resident cells with high resolution, separate between cancer and normal tissue cells, and determine tumours' origin will enrich and expedite broad cancer studies seeking to refine prognostication or cell–cell communication from single-cell transcriptomes.

## Methods

### Defining the pan-cancer tumour microenvironment cell type hierarchy

We defined a structured hierarchy where cell types with transcriptomic similarities are grouped into nodes (Fig. 1a). We first grouped blood cells based on existing relationships that correspond to the hematopoietic hierarchy[48], and kept stromal cells together. For cancer cells, we derived putative groups where transcriptomic similarities are expected based on the cancers' shared organ system, histological subtype, hormonal tissue, or germ layer. These included carcinomas of the digestive system (group 1: colorectal, gastric, esophageal, liver, gallbladder, bile duct, and pancreas), carcinomas not of the digestive system (group 2: lung, breast, prostate, endometrial, and ovarian), and non-carcinomas including soft tissue, neuroendocrine, and nervous system cancers (group 3 cancers: bone, sarcoma, brain, melanoma, and neuroblastoma). We then used a subset of our data, corresponding to one patient sample from each cancer type to evaluate random forest models. Evaluating the proportion of trees voting for each cancer type helped refine the groups and provided guidelines concerning what the subsequent nodes might be.

For example, for the less trivial grouping of kidney cancer, a classification model including all cancers assigned 42.3% of tree votes for kidney, while the majority of the remaining trees voted for various group 2 cancers thus, suggesting linking kidney cancer with group 2. In the other case of lung cancer, we decided to include it in both groups 2 and 3, as separate cancers from both these groups obtained a high number of trees voting for lung. In this case, no clear distinction was observed. In a different case concerning CD8 + T cells, we included CD8 + T cells in both child nodes of the parent node T/NK as different CD8 + T cell populations show more transcriptomic similarities to NK

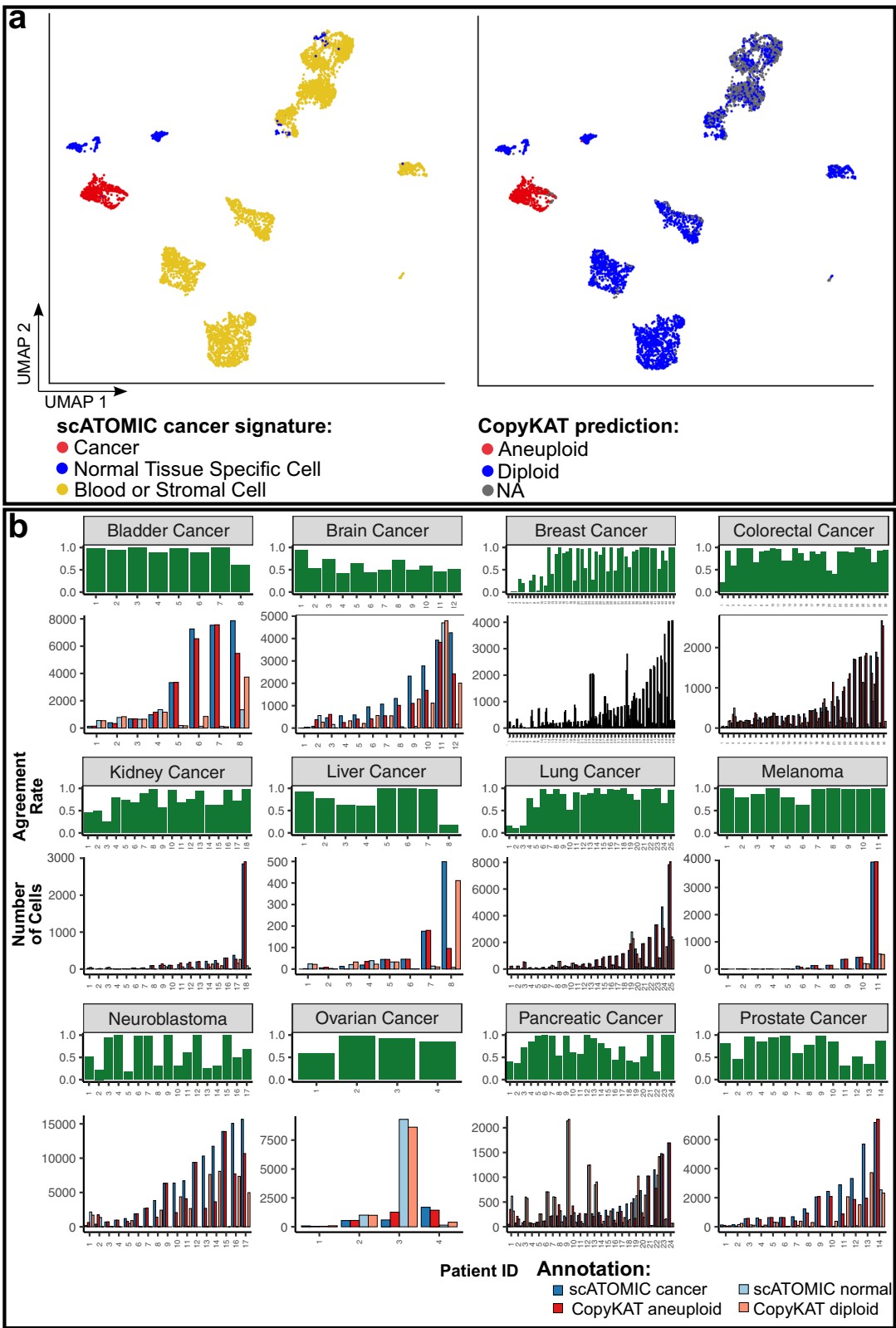

**Fig. 3 | scATOMIC effectively distinguishes between malignant cells and normal tissue specific cells. a** scATOMIC predictions and inferred ploidy in breast cancer patient CID4066[21]. Cells are coloured by scATOMIC predictions and copy number variation (CNV)-based inferred ploidy. scATOMIC-predicted malignant cells are inferred as aneuploid cells while normal tissue cells are inferred as diploid. **b** Comparison of scATOMIC cancer predictions and inferred ploidy statues across the training-independent, external validation datasets. Blue bars represent the number of cells predicted as malignant (solid blue) and non-malignant (transparent blue) by scATOMIC. Red bars represent the number of cells inferred as aneuploid (solid red) and diploid (transparent red). Green bars represent agreement rate in each biopsy. Rates do not include cells without a confident ploidy status (that is received an "NA" annotation by CopyKAT). Source data are provided as a Source Data file.

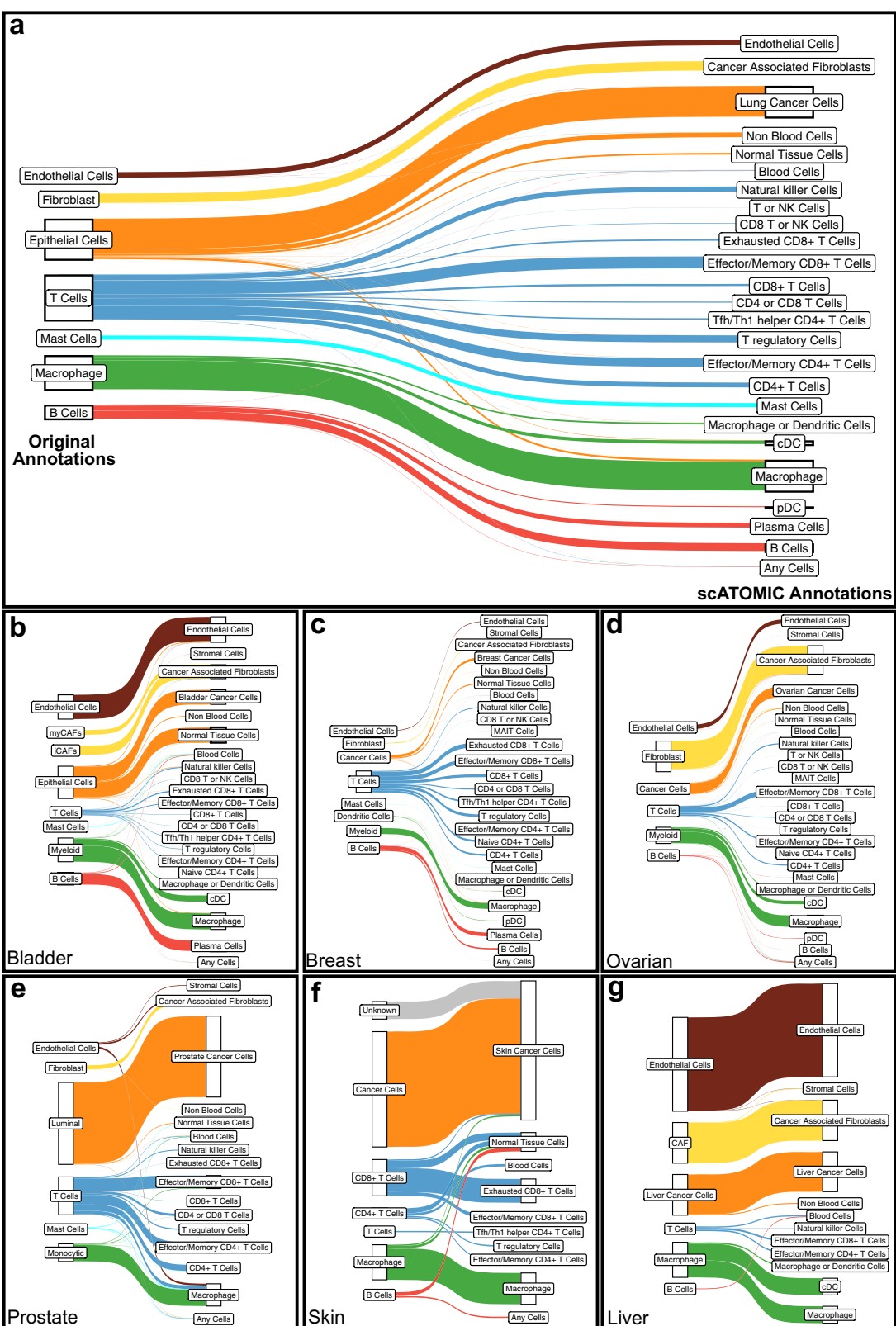

**Fig. 4 | scATOMIC provides greater cellular resolution than original annotations across tumour datasets. a** Sankey plot comparing original cell type annotations to higher resolution scATOMIC annotations in a recent lung cancer biopsy dataset[29]. scATOMIC identifies lung cancer as the tissue of origin and distinguishes these cells from normal lung tissue cells. scATOMIC identifies subtypes of blood cells. **b–g** scATOMIC identifies the tumour origin of common cancers and deliver relatively higher resolution in other cell types[4,33,40–42]. Colours represent the original reported annotations associated with each dataset. The height of each block represents the relative number of cells that received a respective annotation. Source data are provided as a Source Data file.

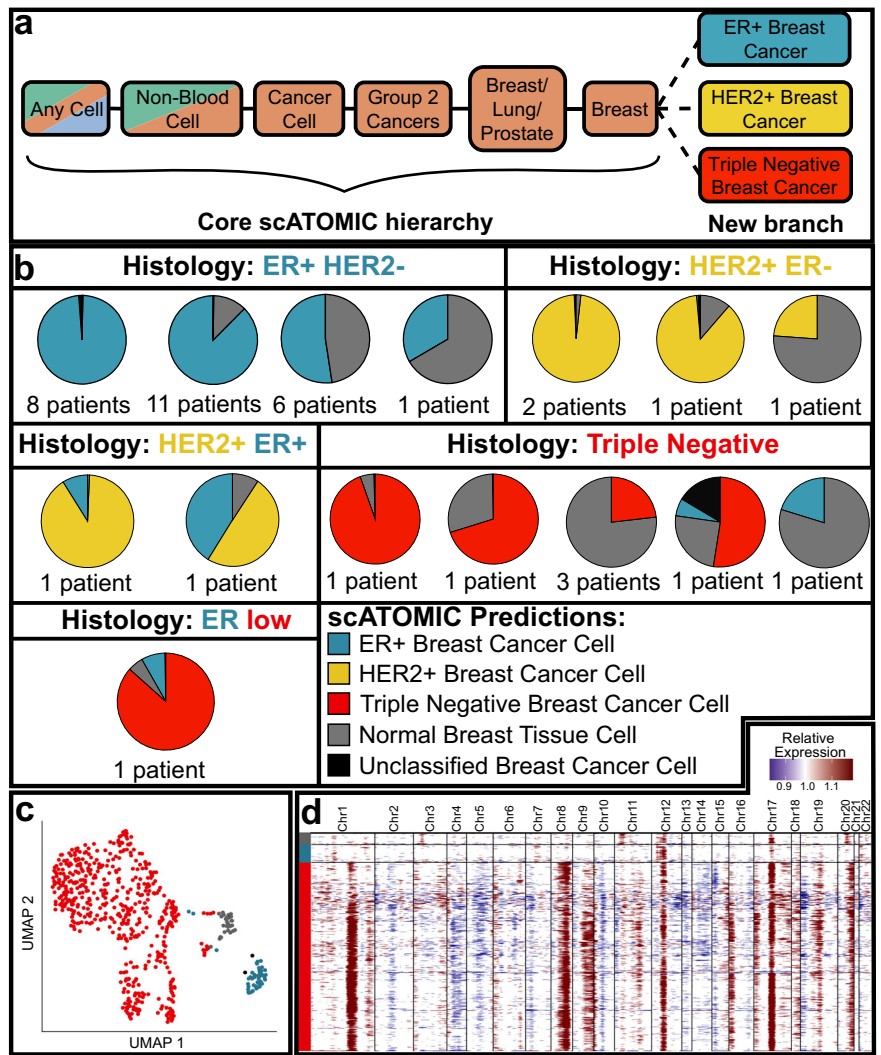

**Fig. 5 | Extending the core scATOMIC model to further classify breast cancer subtypes. a** The terminal breast cancer cell node from the core hierarchy of scATOMIC is extended to subclassify breast cancers into their major ER+, HER2+, and triple negative histological subtypes. **b** Validation of scATOMIC predictions in an external cohort[44]. Pie charts reflecting intra-tumoural breast subtype heterogeneity according to scATOMIC classification are shown for each reported histological subtype. Patient specimens with similar distributions of cell annotations are illustrated together in a single pie chart. **c** Breast cells from an ER-low tumour (Patient:

ER-AH0319) are visualised on UMAP and coloured by scATOMIC predictions. ER + breast cancer cells represent a sub-clonal cancer cell population. **d** Inferred copy number variation (CNV) profiles of cells from ER-low tumour. Red represents inferred gains, while blue represents inferred losses of genomic regions. The y axis is coloured according to scATOMIC prediction. Colours representing scATOMIC predictions apply to all the panels in the figure. Source data are provided as a Source Data file.

(i.e. cytotoxic T cells) while others resemble CD4 + T cells more (Supplementary Fig. 1). We found that this structure yields stronger performance as compared to only having CD8+ cells in one of the branches or generating a single model to differentiate between CD4+, CD8+ and NK all at once. Overall, given that a small random data subset was utilised to infer transcriptomic similarities among classes of cells, it is possible that other hierarchical structures might also be appropriate.

**Feature selection**

For every classification branch within the core scATOMIC hierarchy we selected features as a training input of a random forest model. Raw gene by cell count matrices derived from scRNA-seq analysis were gathered from multiple sources (Supplementary Data 1) and were organized into 24 parent groups (Supplementary Fig. 3). To merge matrices into a particular parent dataset, we removed genes that are not represented in all of the data sources. In each parent dataset we removed cells with <500 expressed genes (as defined by non-zero

counts) or with more than 25% of their reads being mapped to mitochondrial genes.

To find DEGs between each terminal cell type and all other terminal cell types present in the same parent node we used the 'Seurat' R package v4.0.1[16]. Raw gene by cell count matrices were normalised and variance stabilised using the SCTransform function to remove technical variability. Principle components were found using the RunPCA function, on the "SCT" assay. Louvain clustering was performed by first applying the FindNeighbors function on the top 50 PCs, followed by the FindClusters function with a resolution of 2. The identity of the resulting cell clusters was determined by the transcriptomic-independent ground truth associated with the training datasets. For each model, DEGs were found using the FindMarkers function (two-sided Wilcoxon rank sum test) with ident.1 set to include all clusters containing a particular terminal cell type and ident.2 being all other cells in the parent node. The function returned a list of DEGs per class that passed default Seurat filtering settings: a $\log_2$ fold-change of at least 0.25, and at least 10% of the cells in ident.1 or ident.2

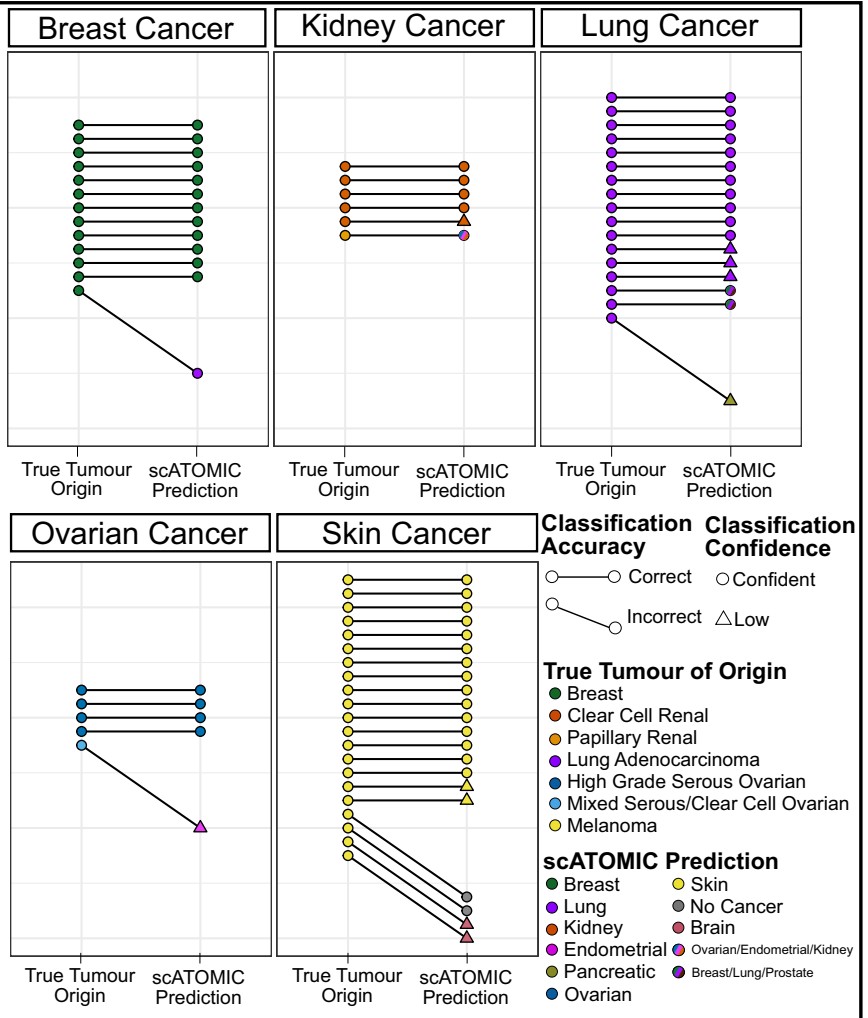

**Fig. 6 | scATOMIC accurately identifies the tissue of origin in metastatic tumour biopsies.** scATOMIC was applied to 62 metastatic tumours from breast, kidney, lung, ovarian and skin. Metastatic sites included the brain, lungs, GI tract, liver, adrenal glands, lymph nodes, abdomen, and peritoneal cavity. Each pair of dots represents the true tumour origin and the predicted origin. Horizontal connected lines represent correct predictions, while diagonal lines represent incorrect predictions. True tumour origins are coloured by the reported cancer subtype. Circular points represent confident annotations, while triangular points represent low-confidence annotations (Methods). Multi-coloured points represent tumours that received an intermediate scATOMIC annotation. Source data are provided as a Source Data file.

expressing the respective gene. We defined a differential expression score (DES) as the difference of the fraction of cells expressing a non-zero value for a respective DEG in ident.1 and ident.2 (DES = pct_expr_ident.1 – pct_expr_ident.2) to capture DEGs more specifically expressed in any particular cell type. For each terminal cell type, we kept genes with a DES greater than the mean DES of all DEGs for that cell type. We removed all ribosomal genes. We also removed DEGs that had a pct_expr_ident.2 >40% to ensure high performance when interrogating datasets with large technical variation. For the same reason, we set a minimum and maximum number of DEGs for each cell type at 50 and 200. Specifically, a minimum number of features was set to mitigate potential issues in classifying cells with high levels of technical dropout. In the case where there are fewer than 50 DEGs with DES higher than the mean, we kept the top 50 DEGs ranked by DES. Features that were used for each cell class at each classification layer are provided in Supplementary Data 7.

### Random forest modeling
For each classification branch within the core scATOMIC hierarchy, we trained a random forest model on cells within a respective parental node and features selected as described above. To minimise bias associated with imbalanced classification towards majority classes[49], we randomly sampled an equal number of cells from each terminal class, with replacement. Library size of each single cell was normalised by using the library.size.normalise function from the 'Rmagic' v2.0.3 package[50]. Prior to training each model, normalised counts were filtered to include selected features and cells within the corresponding parental node. Read count values were transformed to a fraction of the total filtered counts. A random forest classifier was trained on the transformed matrix using the 'randomForest' R package v4.6–14 with 500 trees and default parameters[51]. Each random forest was trained to classify the terminal nodes present within the corresponding parental node. The specific cell type organization of the 24 classifiers is detailed in Supplementary Fig. 3.

### Applying scATOMIC to query datasets
Before using scATOMIC on query datasets, the interrogated data was processed as follows. Raw gene by cell count matrices were filtered to remove cells with non-zero counts for <500 genes or with more than 25% of their reads being mapped to mitochondrial genes. We imputed missing values in DEG using the *magic* function from the 'Rmagic' package[50], where all the cells within the query dataset act as a

reference. In datasets where there were no reported values across all the samples for specific selected features, we assigned a value of zero before imputation. Following each classification task, every cell received a vector of prediction scores corresponding to the percentage of trees voting for each terminal class in the random forest model. The values in each vector within the next immediate parent node were then summed to generate IGSs. For example, when the first model interrogating all the terminal classes in the hierarchy ran (i.e. the parent model "Any Cell"), the output for each cell was composed of two intermediate group scores. The first corresponded to the sum of trees voting to all the terminal cell classes belonging to the "Blood Cell" parent node and the second IGS for the "Non-Blood Cells" parent node (Fig. 1d). Data from all the cells in the interrogated sample was used to derive IGS parent distributions. Cells which received an IGS greater than the defined parent threshold continued down the classification hierarchy until they were terminally classified (Fig. 1e). At any stage, if a cell received an IGS lower than the calculated threshold it was annotated based on the previous parental node (a less specific classification). An IGS threshold for a classification to be deemed confident was automatically determined (Supplementary Fig. 4). Using the ajus function from the 'agrmt' package v1.42.4[52], IGS distributions for each IGS calculated among all cells within a parental node are classified as either unimodal or bimodal. Unimodal distributions suggest a layer includes one subtype, while bimodal distributions indicate there is likely more than one subtype. For unimodal distributions we set the IGS threshold to be the mean IGS ($\mu$) – 3 standard deviations ($\sigma$). For bimodal distributions, using the em function from the 'Cutoff' R package[53] v0.1.0, we fit a mixture model for the distributions and predicted estimates of mean and standard deviations for both distributions using the expectation maximization algorithm. We selected a conservative approach when a mixed cell type population exists in a layer by setting the IGS threshold in bimodal distributions to be the mean of the higher score modality ($\mu_2$) – 2 standard deviations ($\sigma_2$). We set the maximum IGS threshold to be 0.7, as in some distributions, such as when a single pure population remained for classification, unreasonably high thresholds may be obtained.

A schematic and description of the main functions is detailed in Supplementary Note 2.

## Flagging cells with lower confidence annotations
To provide a way for scATOMIC users to evaluate the confidence of their cell annotation output, we devised a secondary post-classification flag to warn about low-confidence annotations. To define low-confidence annotations, we used the results obtained by external validation (Fig. 2b). For every model throughout the hierarchy, we determined the median IGSs (or PS for terminal nodes) across samples for correctly annotated cells ($X$) and incorrectly annotated cells ($Y$). Correct versus incorrect status was defined by the terminal annotation of single cells. For example, in terminally annotated breast cancer cells, median $IGS_{non-blood}$, $IGS_{non-stromal}$, $IGS_{group2-cancer}$, $IGS_{breast/lung/prostate}$, $PS_{breast}$ for all the correctly and incorrectly annotated cells in the validation data were recorded. We derived confidence thresholds based on the overlap between the distributions of $X$ and $Y$ using their quartiles (Q). When there was low overlap (defined as minimum $X > Q3$ $Y$), the threshold was set to the minimum $X$. When there was intermediate overlap (defined as minimum $X < Q3$ $Y$, yet $Q1$ $X > Q3$ $Y$), the threshold was set to Q3 $Y$. In all remaining cases where there was high overlap (defined as $Q1$ $X < Q3$ $Y$), segregation could not be made and the classifications of query cells in such cases were deemed confident. For example, all CD4 + T cells that are misclassified as CD8 + T cells will still obtain comparable $IGS_{blood}$ with respect to correctly classified CD8 + T cells, both being subtypes of blood cells. If at any model throughout the hierarchy a low-confident IGS is observed, a flag is applied. In addition, we assigned a sample-level confidence metric derived from the proportion of cells receiving a confident annotation

based on this flag. To maximise identification of potentially poor-quality samples, in any new interrogated sample, if <75% of cells receive confident annotations, a warning will be issued (Supplementary Fig. 12).

## Scoring cancer signatures to refine cancer cell predictions and identity of normal tissue-specific cells
To identify potential normal tissue specific cells that are not defined in the core scATOMIC TME hierarchy, we employed a post classification method for scoring cancer specific modules in each cell. Lists of genes differentially expressed between different cancer types and their matched normal tissues were obtained from OncoDB[31]. We selected DEGs from OncoDB[31] with a reported log2 fold change >1 or <−1 and an adjusted $P$ value <0.01. Since OncoDB[31] is based on bulk RNA-seq, we further filtered the DEG list to only include those with reported expression values in the query scRNA-seq dataset. Upregulated gene programs and downregulated gene programs[31] were scored using the AddModuleScore function from Seurat[11,16] in each cell predicted as cancer by random forests. Ward.D2 hierarchical clustering was then performed on a Euclidean distance matrix of each cell's upregulated and downregulated cancer programs. Two groups were derived using the cutree function. At this stage, scATOMIC evaluates the percentage of normal cells in each group corresponding to those cells annotated as either blood, stroma or cancer cells with lower upregulated program scores compared to downregulated program scores. As the AddModuleScore function uses average expression of control feature sets across all cells in the dataset, the calculated scores are affected by the proportion of normal cells present. Thus, we filtered out all confident normal cells in the cluster with a greater percentage of normal cells and repeated the AddModuleScore pipeline to identify additional normal cells that were overlooked in the first iteration. We repeated scoring of cancer programs, hierarchical clustering, calculating the percentage of normal cells and filtering until both clusters contained no more than 20% normal cells. Cells that were initially classified as cancer which were scored as normal cells were given a normal tissue cell label.

## Benchmarking scATOMIC
We validated scATOMIC's performance using internal cross validation and external validation datasets. Internal validation was performed by splitting the training dataset (Supplementary Data 1) into five subsets containing equal proportions of each cell type. For each iteration we used 4 subsets as scATOMIC training dataset and applied scATOMIC to the held out independent test subset. F1 scores were calculated for each terminal cell type at each iteration (Supplementary Data 1). AXL+ dendritic cells (ASDC) represent a rare, recently discovered, transitional state between cDCs and pDCs[54]. Due to their small numbers in the training data, these cells were omitted from the internal validation procedure.

For external validation of scATOMIC we used 424,534 cancer, blood and stromal cells gathered from various sources (Supplementary Data 3). For ground truth, we relied on the authors' annotations of stromal cells. To improve reliability, cells that were annotated as cancer by the authors were subjected to additional validation by CNV inference using CopyKAT[19]. For blood cells, we used CITE-seq data with protein surface markers supporting the author derived cell type annotations. We excluded cell types from individual samples if their number was <30. Per sample F1 scores were calculated for each terminal cell type. During both scATOMIC's internal and external validation processes (Fig. 2a, b), we considered correct intermediate classifications as true positives.

## Comparison between scATOMIC and other tools
We compared scATOMIC's performance to existing scRNA-seq classifiers. In this comparison only, we bypassed the use of IGS cut-offs in

scATOMIC to enable comparison to other tools that cannot output intermediate cell classes. Of note, a separate analysis evaluating scATOMIC's performance under this forced mode against its default (unforced mode) favoured the latter by indicating that the use of intermediate annotations more frequently avoids derivation of incorrect terminal annotations than restricting the output of correct terminal classes. The same training and validation datasets provided to scATOMIC were also used to provide a comparison of scATOMIC with the performance of all the other tested tools. A SingleCellNet[36] random forest model was trained on a balanced reference of 2500 random samples of each cell type, using default parameters. SingleCellNet expects a class-balanced matrix as input. We provided the same class-balanced matrix used in scATOMIC's first classification model, representing all the cell classes. CHETAH[37] is using an alternative hierarchical classification approach. We used CHETAH with its default settings with the exception of the 'thresh' parameter that was set to zero to enforce terminal annotations, similar to scATOMIC. Scmap-cell[17] classification was performed using default parameters. To enable SingleR processing of the large reference dataset, we applied its pseudobulk implementation by setting 'aggr.ref' to TRUE[15]. For the comparison with Seurat[16,35], we partitioned the pan-cancer TME training reference according to batch and applied the reciprocal PCA workflow. We transferred labels to query samples using the TransferData function. For the comparison with scType[38], we provided scType with the list of features corresponding to each terminal class in scATOMIC's reference which were derived in the first classification node (Supplementary Data 7). Otherwise, default parameters were used. To compare scATOMIC's final cancer cell prediction with the CNV inference approach of detecting malignant cells we used CopyKAT[19] with its default settings (Fig. 3). Both aneuploid and diploid cells from the external validation biopsies were included in this analysis. Agreement rate was defined as the simple percentage agreement using the agree function from the irr[55] R package. Cells which received an NA ploidy annotation were omitted from the calculation.

### Analysis of run time and memory usage
We applied each tool without parallel processing, as only some tools (scATOMIC, Seurat, SingleR) provide that functionality. We considered the time and memory usage for query datasets following model training as some methods bypass this step and rely on a given cell marker list (scType). Using the peakRAM[56] R package we monitored the time for classification and maximum RAM used. We compared the time and RAM required for the classification of cell types prior to cancer signature scoring by scATOMIC to SingleR, Seurat, scmap-cell, SingleCellNet, CHETAH, and scType final classifications. As appropriate, we compared the time and memory for the cancer signature scoring step that differentiates cancer from normal tissue cells to CopyKAT.

### In silico dilution assay
For each primary tumour sample in the external validation, we ran scATOMIC and CopyKAT on all non-malignant cells while iteratively reducing the number of malignant cells. Specifically, we sampled 10–100 malignant cells in increments of 10. Only cells that both scATOMIC annotated as cancer cells and CopyKAT inferred as aneuploid were sampled.

### Breast cancer subclassification
We extended the core scATOMIC model to subclassify breast cancer cells into their histological subtypes. The model extending breast cancer into its molecular subtypes was trained using a combination of two datasets. The first included the breast cancer cell lines data within the core training dataset (Supplementary Data 1, 2), where the assigned molecular subtypes were based on annotations found in the Cancer

Cell Line Encyclopedia DepMap portal[57] and the second, primary tumour data from Wu et al.[21] with immunohistochemistry information (Supplementary Data 4). We used the clinical molecular subtypes of HER2 + , ER + and triple negative as classes. There was not sufficient data available to train and test HER2 +/ER + and HER2-/ER + as separated classes. We evaluated this extension's performance on an external set of 40 tumours from Pal et al.[44] (Supplementary Data 5). One tumour containing fewer than 100 cancer cells was omitted from this analysis. The ground truth of ER and HER2 status in the testing set was received by correspondence with the authors.

### Visualising inferred CNVs
To visualise the inferred CNV profile in the ER-low tumour we used the 'inferCNV'[58] package with the cutoff variable set to 0.1 and all other variables set to default. We defined normal reference cells as cells annotated by scATOMIC as blood or stromal cells.

### Applying scATOMIC to metastatic tumours
scATOMIC was applied to predict the tumour of origin in 62 metastatic tumours. Individual samples used are described in detail in Supplementary Data 6. We defined the scATOMIC tumour of origin prediction by taking the cancer type called in the majority of cancer cells in the sample.

### Reporting summary
Further information on research design is available in the Nature Portfolio Reporting Summary linked to this article.

## Data availability
All the scRNA-seq data used in this work are publicly available. Datasets retrieved from the Gene Expression Omnibus can be downloaded using the following accession numbers: GSE164378[16], GSE118257[24], GSE114374[25], GSE135893[27], GSE140819[29], GSE148673[19], GSE176078[21], GSE132465[22], GSE125449[41], GSE131907[23], GSE115978[33], GSE137804[59], GSE141445[42], GSE154826[60], GSE123139[34], GSE161529[44]. Datasets retrieved from the Broad Institute Single Cell Portal are available with the following accession numbers: SCP542[20], SCP1288[61], SCP1415[32]. The remaining datasets were downloaded directly from links provided in their corresponding publications: Madissoon et al.[26] [https://data.humancellatlas.org/explore/projects/c4077b3c-5c98-4d26-a614-246d12c2e5d7], Chen et al.[4] [https://static-content.springer.com/esm/art%3A10.1038%2Fs41467-020-18916-5/MediaObjects/41467_2020_18916_MOESM2_ESM.zip], Couturier et al.[62] [https://datahub-262-c54.p.genap.ca/GBM_paper_data/GBM_cellranger_matrix.tar.gz], Qian et al.[40] [https://lambrechtslab.sites.vib.be/en/pan-cancer-blueprint-tumour-microenvironment-0], Young et al.[63] [https://www.science.org/doi/suppl/10.1126/science.aat1699/suppl_file/aat1699_datas1.gz.zip], Peng et al.[64] [https://zenodo.org/record/3969339], Zheng et al.[45] [https://zenodo.org/record/5461803]. Additional descriptions of these datasets are provided in Supplementary Data 8. All re-processed data used for training and validation have been deposited in Zenodo and are available through the following link: https://doi.org/10.5281/zenodo.7419236[65]. Bulk RNA sequencing cancer specific signatures were obtained from OncoDB[31] [https://oncodb.org/data_download.html]. Subtypes of cancer cell lines were derived from the Cancer Cell Line Encyclopedia in DepMap[57] [https://depmap.org/portal/ccle/]. Source data are provided with this paper.

## Code availability
The scATOMIC R package, associated code and user manual are available at the abelson-lab/scATOMIC GitHub repository: https://github.com/abelson-lab/scATOMIC[66]. Additional scripts to reproduce the figures in the manuscript are deposited in Zenodo and are available through the following link: https://doi.org/10.5281/zenodo.7419236[65].

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

## Acknowledgements

This work was supported by a grant from The Banting Research Foundation (Discovery Award #2021-1418) and Investigator Awards received from the Ontario Institute for Cancer Research with funds from the province of Ontario to S.A. and P.A. I.N.M. obtained funds from the Ontario Graduate Scholarship Program – University of Toronto. We thank Salman Basrai for his assistance checking the validity and accuracy of the scATOMIC code, as well as on his comments and suggestions concerning the package documentation.

## Author contributions

All authors discussed the results, wrote, and commented on the paper. I.N.M. developed the concept, performed data analysis, developed the tool, and wrote the code. D.S. contributed to statistical analysis and model development. S.A. and P.A. conceived the idea, contributed to data analysis, led, and supervised all aspects of the study.

## Competing interests

The authors declare no competing interests.
