## [Peer review file · Nature Communications]

REVIEWER COMMENTS

Reviewer #1 (Remarks to the Author): Expert in single-cell RNA-seq, cancer genomics, tumour microenvironment, and bioinformatics

The manuscript by Nofech-Mozes et al. presents a novel automatic cell annotation tool for scRNA-seq data. The authors pointed out critical computational problems encountered in clustering and annotating, such as inter-patient heterogeneity, overlapping expression profiling among specific cell types, among others. To tackle these difficulties, the authors designed the scATOMIC algorithm which adopts a hierarchical classification strategy, and trains random forest-based models with inputs of features derived from scRNA-seq reference data. In the benchmark datasets, scATOMIC was shown to have great effectiveness to accurately annotate cells from query datasets and particularly outperform multiple methods in terms of the correct identification of tumor cells. Overall, the algorithm design is rational and the manuscript is well-written. However, the achieved classification of scATOMIC is relatively coarse-grained, limiting its applications. Below listed are the concerns that need to address or clarify:

- 1) In view of scATOMIC being designed for dissecting the tumor microenvironment, the authors should take into consideration the tissue specificity for the construction of training references. The authors used the peripheral blood datasets as the reference for immune cells and the normal tissue datasets as the reference for stromal cells, which seems unreasonable. Immune cell composition in the blood is very different from that infiltrated into the tissues. For example, monocytes are enriched in the blood, while macrophages are exclusively in the tissues. Even those stromal cells, the tumor-associated fibroblasts (CAFs) exhibit distinct phenotypes from the normal tissues. This also explains why mast cells, a common cell type in the tumor specimens, were not identified in the benchmarking (Line 183). The authors should use available cancer datasets as references and re-train the models.
- 2) The currently used reference datasets were generated by several different platforms, such as 10X and CITE-seq, and the future input query datasets may be also generated by other platforms. The authors should address whether the batch effects caused by different platforms affect the model building and limit the querying input.
- 3) It is not clear to this reviewer how scATOMIC handles the case when the cell type corresponding to the query cells is not in the training reference.
- 4) For the benchmarking section, the authors should use other independent testing datasets instead of the training set of scATOMIC. Furthermore, the existing methods used for comparison are relatively old,

and the authors should consider using some newly published methods to benchmark the performance of scATOMIC.

5) Line 173: the section titled “scATOMIC annotations increase cellular resolution in tumour biopsies” seems inappropriate. The classification achieved by scATOMIC is still relatively coarse-grained, while cell types such as pDC and mast cells may be also easily identified using traditional unsupervised clustering in conjunction with manually inspecting the expression of canonical marker genes. Furthermore, it is not clear whether scATOMIC still predict accurately if the model was trained for fine-grained cell types (like different T cell states).

Reviewer #2 (Remarks to the Author): Expert in single-cell RNA-seq, bioinformatics method development, and statistics

The authors present scATOMIC an automatic-annotation tool for single-cell RNAseq datasets derived from human cancer samples. The tool uses a novel hierarchical classification approach which they name "reversed hierarchical classification and repetitive elimination of parental nodes" which is based upon randomForest models trained at each node of a pre-determined cellular hierarchy. They train their method on a set of cancer cell-lines and then benchmark it against current state-of-the-art annotator on some flow-sorted data as well as compare it's ability to identify malignant cells to using CNV analysis. They demonstrate both that scATOMIC correctly annotates cells within its a priori defined hierarchy and that it will not incorrectly annotate cells with identities outside of the pre-defined hierarchy. Finally they present a use case for predicting the tumour-type of origin from metastatic cancer cells. Overall, it seems like an interesting and useful approach. I successfully installed the software and ran the tutorial.

Major Concerns

1. The origin of the cellular hierarchy that is fundamental to the construction and operation of the method is unclear. Certain seemingly unrelated cancers are grouped together (e.g. kidney cancer grouped with Ovarian rather than Hepatobilliary and Pancreatic) , some leaves are specific cell-types while others are hybrid cell-type (e.g. Fibroblast/SmoothMuscleCell). Likewise distinct branches converge for CD8+ T cells which does not make sense give the logic of the approach. Could the authors clarify why they constructed the “tree” in this way and include in the manuscript guidance to users on how to construct their own trees for the application of expanding the method to additional cell-types.

2. It is unclear how scATOMIC compares to SingleR, while the authors present SingleR as vastly inferior to scATOMIC in their benchmarking (Figure 2), but then show scATOMIC to perform nearly identically to SingleR in Figure 4.

3. During the benchmarking the authors force scATOMIC to report terminal leaf annotations, however, in normal use the method would report internal nodes. What proportion of the time would the incorrect terminal annotations be reported by scATOMIC vs a correct though imprecise internal node? How often does scATOMIC report an internal node instead of the correct terminal node? Should users force scATOMIC to return terminal nodes for the best annotations?

4. Extended Data Figure 1 does not clearly support the conclusions for which it is cited in the manuscript. It shows several marker genes that can easily distinguish CD8+T, CD4+T, and NK cells as well as the fact that CD8+T uniquely express both genes expressed in CD4+T and genes expressed in NK cells. Yet the authors claim it shows that it is difficult to distinguish these cell-types based on marker genes.

5. Extended Data Figure 2 does not clearly support the conclusions for which it is cited in the manuscript. Panel c/d does seem to be somewhat clustered by cancer type - particularly the lower right corner (pink), lower left corner (yellowy-green) and top right corner (yellow-y orange). And panel e uses general biological processes vaguely related to cancer (some which are more related to the immune response to cancer not the cancer cells themselves - e.g. IFNresponse) not transcriptional programs derived from comparing malignant cancer cells to healthy control cells. Furthermore, despite claiming citing this figure that geneset signatures do not distinguish cancer cells and cancer cell-types they then use geneset signatures (from OncoDB) in the final stage of scATOMIC. If they used the geneset signatures from OncoDB in this figure would they see the same results?

6. The authors claim: "inter-patient variability as a major obstacle for accurate classification" when considering their cell-line data. However, these cancer cell-lines have been grown for many generations in the lab, they are not primary cancer cells from patients. Thus they cannot conclude differences between the cell-lines result from inter-patient variability as genetic and transcriptional changes resulting from immortalization and in-vitro evolution are equally likely to cause the observed variability in expression profiles.

7. In the main text the authors claim: "RHC-REP prioritizes the features with the highest specificity to the interrogated cell types based on their absolute expression (Methods, Fig. 1b)" However, upon reading the Methods section I found that they identified features using differential expression testing of the SCTransformed gene expression values using the default wilcox test which in fact uses purely the relative rank of normalized expression values, not their absolute raw counts.

Minor Concerns

8. The cell-type resolution of scATOMIC is quite low. In my experience identifying CD8+ T cells, CD4+ T cells and NK cells is simple to do and generally the type of cancer a dataset originates from is known a priori. The application to identifying malignant cells is more interesting and should be highlighted in the paper more.

9. In Figure 3b it is completely inappropriate to use a line graph. Line graphs imply relationships between adjacent points and in this case adjacent points derive from independent tumour samples.

Reviewer #3 (Remarks to the Author): Expert in single-cell RNA-seq, bioinformatics, statistics, and tumour immune microenvironment

This manuscript introduces a new computational tool scATOMIC for classifying cells in tumor single-cell RNA-seq samples. Compared to existing tools, scATOMIC can more comprehensively annotate both malignant cells and non-malignant cells including immune and stromal cells. Through analysis of data from multiple tumor types, the authors demonstrated the ability of scATOMIC to better annotate cells in tumor samples with increased accuracy and resolution, and to infer tumor origins of metastatic samples. Overall, the manuscript is clearly written, and scATOMIC can be a useful tool for annotating cells in tumor single-cell RNA-seq samples. Below are several issues I suggest the authors to address to make the study more solid and useful.

Major issues:

1. Data from different single-cell RNA-seq platforms and protocols have different characteristics. It is unclear whether the training and test data in this study are all from the same platform. If so, which platform? If not, the authors should provide the platform/protocol information for each dataset. In real applications, user's query data may be generated by a platform different from the one used to train the scATOMIC model. Can scATOMIC still produce reliable cell annotation in that scenario? How does the classification performance in query data vary across different platforms? A systematic benchmark to answer this question will be important.

2. The performance shown in this manuscript is on the cell-level. Does the cell classification accuracy vary a lot across samples? How much variation in performance should one expect across samples? Table S3 seems to suggest that the cell classification accuracy is very low in some samples. In real applications, how can users decide whether they should trust the predicted cell labels in their query sample (i.e. whether each query sample has high or low prediction accuracy)? Is there an objective statistic to guide this decision?

3. It is unclear what Extended Data Fig. 1 is showing. There is no description of the analysis procedure, making the data here hard to understand and reproduce.

4. Figure 4 shows that scATOMIC can annotate cells with higher resolution compared to other annotations. A careful examination of this figure and Figure 1 suggests that the increased resolution is mainly for immune cells, and the resolution for cancer cell subtypes within each cancer type is still quite low. This seems to be a limitation of the current scATOMIC that should be discussed. It could be even better if the authors can systematically address this in a wide variety of cancer types like how they further classified breast cancer cell subtypes.

5. For the breast cancer analysis, the authors claim that "scATOMIC correctly subtyped 39 of the 40" biopsies. It is unclear how these statistics are obtained. In Table S5, there are 40 samples, and both of the HER2+/ER+ samples are incorrectly predicted as HER2+ instead of HER2+/ER+. This means the correctly subtyped biopsies have to be no more than 38. Where does the number 39 come from?

6. It could be interesting to discuss or explore with data to show how the improved annotation of malignant and non-malignant cell types would allow one to better study interactions between different cell types (e.g. tumor cells and immune cells) in the TME.

Reviewer #4 (Remarks to the Author): Expert in cancer genomics, bioinformatics, and machine learning

Nofech-Mozes et al. proposed a tool called scATOMIC for the annotation of malignant and non-malignant cells. The authors included a breast cancer dataset as a validation cohort with clinical relevance.

This is a highly relevant and well-conducted study and the paper is well-written. However, there is a lack of information on the manuscript that must be addressed before publication.

Major:

1. The authors have to provide a schematic figure of the tool including the main function and the inputs-outputs.
2. Did the authors check the performance of the tool before and after batch effect correction? In the manuscript, they mentioned the intra-patient similarities (“suggesting inter-patient variability as a major obstacle for accurate classification”) so I am wondering if the performance of the tool can be improved using harmony with Patient ID as the cofactor.
3. Regarding the comparison with other cell-type classification methods the authors have to present also the tool's computational time and the complexity of the proposed tool analysis for every module.
4. The GitHub repository is very well structured and is ensured reproducibility. However, the data should be under a stable DOI (e.g. Zenodo). Moreover, a short description of the main functions is necessary to be described also to the manuscript (e.g. as supplementary material)
5. The study will gain a lot of power in their approach if the authors include as an additional clinical scenario the measurable residual disease after treatment.

Minor:

1. line 42: has to have
2. line 93: remove As
3. The link to the GitHub repo does not work
4. In figure 5, the authors have to include the number of cells per patient and the predicted number of cells in each case

Reviewer #1 (Remarks to the Author): Expert in single-cell RNA-seq, cancer genomics, tumour microenvironment, and bioinformatics

The manuscript by Nofech-Mozes et al. presents a novel automatic cell annotation tool for scRNA-seq data. The authors pointed out critical computational problems encountered in clustering and annotating, such as inter-patient heterogeneity, overlapping expression profiling among specific cell types, among others. To tackle these difficulties, the authors designed the scATOMIC algorithm which adopts a hierarchical classification strategy, and trains random forest-based models with inputs of features derived from scRNA-seq reference data. In the benchmark datasets, scATOMIC was shown to have great effectiveness to accurately annotate cells from query datasets and particularly outperform multiple methods in terms of the correct identification of tumor cells. Overall, the algorithm design is rational and the manuscript is well-written. However, the achieved classification of scATOMIC is relatively coarse-grained, limiting its applications. Below listed are the concerns that need to address or clarify:

We thank the reviewer for his comments and the time taken to review our manuscript. Responses are indicated in red, while references to the manuscript are shown in blue.

1) In view of scATOMIC being designed for dissecting the tumor microenvironment, the authors should take into consideration the tissue specificity for the construction of training references. The authors used the peripheral blood datasets as the reference for immune cells and the normal tissue datasets as the reference for stromal cells, which seems unreasonable. Immune cell composition in the blood is very different from that infiltrated into the tissues. For example, monocytes are enriched in the blood, while macrophages are exclusively in the tissues. Even those stromal cells, the tumor-associated fibroblasts (CAFs) exhibit distinct phenotypes from the normal tissues. This also explains why mast cells, a common cell type in the tumor specimens, were not identified in the benchmarking (Line 183). The authors should use available cancer datasets as references and re-train the models.

We agree with the comment that PBMCs and normal-tissue derived stromal cells may exhibit distinct phenotypes which are different from cells infiltrating tumour microenvironments. Moreover, we appreciate that those differences may impact classification performance. We originally used only PBMC data which is linked to transcriptome-independent, gold standard cell type annotations via CITE-seq. Using such high-quality data concerning cell annotations made a well-founded case for accurate model training, validation, and benchmarking.

To address the reviewer's comment and enable a proper integration of cancer datasets in the training set, we conducted an independent analysis to filter out cells with author-derived annotations that might be less reliable (**Fig. R1**). We re-trained the models with tumour-derived training data, including classes corresponding to tumour-infiltrating lymphocyte and stromal cells (**Supplementary Table 1**). In addition, we gathered enough high-quality data to train and test a new class for mast cells and a classification layer distinguishing macrophages from monocytes. Most importantly, all the new data used for external validation and benchmarking corresponds to cancer biopsies and tumours infiltrating cells (**Supplementary Table 3,5**). **Fig. 2b-c** illustrates the re-trained models' high level of performance across these cancer datasets where all cells used include high confidence annotations via the above-mentioned filtering, CITE-seq, or inferred CNVs in cancer cells.

Figure R1. Filtering cells with inconclusive manual annotations. Here we highlight original annotations of different cells provided by authors and show as an example that (a) following unsupervised clustering “endothelial cells” fell into three separated clusters, two of which were dominated by other cells, either “Fibroblasts” or “Monocytic”. (b) Endothelial cells that dominated a single largest cluster were kept while the other cells originally designated as “Endothelial” were omitted from training. The same strategy was used for the other cell types, that is, the one cell class dominated a cluster was kept. In addition, the identities of clusters were confirmed using the expression of cell type specific canonical markers.

2) The currently used reference datasets were generated by several different platforms, such as 10X and CITE-seq, and the future input query datasets may be also generated by other platforms. The authors should address whether the batch effects caused by different platforms affect the model building and limit the querying input.

We added a new column in Supplementary Table 8 highlighting which platform(s) was used for each dataset in our study. Data curation was unbiased therefore, it is evident that 10X is the most commonly used platform for cancer single-cell transcriptomic studies to date. Some variations can be seen in the type of gene expression libraries, with the majority associated with the 3’ end of transcripts. Although more rare, we found and included public data from other lower-throughput platforms during model building. Due to this bias towards 10x and data rarity from other platforms, building accurate models that are platform-specific is currently problematic. Nonetheless, we now provide a direct evaluation of scATOMIC performance across 10X scRNAseq, single nucleus RNA sequencing, Smart-seq2 and MARS-seq datasets available for melanoma.

We now address scATOMIC performance across different training batches and scRNA-seq platforms in the main text (lines 125-130):

“To ensure scATOMIC was not heavily impacted by batch effects, we also trained 4 scATOMIC iterations, where all cells from intact technical batches were held out from training. We found no significant difference in the performance of the models on the held out batches (Extended Data Fig. 5a). We further tested scATOMIC performance across different scRNA platforms using external

melanoma datasets^{29,32-34} and again, found no significant difference in F1 scores (Extended Data Fig. 5b).”

The results, included in a new supplementary figure (Extended Data Fig. 5), indicate no significant difference in performance using these datasets. We observed consistently high F1 scores across blood, stroma and cancer and across platforms. In addition, we would like to clarify that the CITE-seq data used in our study was generated using the 10X platform (Now mentioned in Supplementary Table 8). In these cases, the information concerning gene expression was used for either training or testing, while the antibody-bound synthetic oligos that were sequenced served as a transcriptome-independent confirmation of cells’ identities.

3) It is not clear to this reviewer how scATOMIC handles the case when the cell type corresponding to the query cells is not in the training reference.

We want to refer the reviewer to the new Supplementary Note 1 (“Query of unrepresented cell types”). In this document we illustrate in detail how cells that receive an IGS lower than the automatically derived confidence cutoff at intermediate or terminal classification nodes maintained less informative, yet correct classifications. This is expected to happen more frequently in cases where cells are not in the reference.

In addition, we embedded a new flag to let users know of lower confidence classifications. We provide additional details concerning this new flag in the Methods section (lines 407-424):

“Flagging cells with lower confidence annotations

To provide a way for scATOMIC users to evaluate the confidence of their cell annotation output, we devised a secondary post-classification flag to warn about low-confidence annotations. To define low-confidence annotations, we used the results obtained by external validation (Fig. 2b). For every model throughout the hierarchy, we determined the median IGSs (or PS for terminal nodes) across samples for correctly annotated cells (X) and incorrectly annotated cells (Y). Correct versus incorrect status was defined by the terminal annotation of single cells. For example, in terminally annotated breast cancer cells, median $IGS_{non-blood}$, $IGS_{non-stromal}$, $IGS_{group2-cancer}$, $IGS_{breast/lung/prostate}$, PS_{breast} for all the correctly and incorrectly annotated cells in the validation data were recorded. We derived confidence thresholds based on the overlap between the distributions of X and Y using their quartiles (Q). When there was low overlap (defined as minimum $X > Q3 Y$), the threshold was set to the minimum X. When there was intermediate overlap (defined as minimum $X < Q3 Y$, yet $Q1 X > Q3 Y$), the threshold was set to $Q3 Y$. In all remaining cases where there was high overlap (defined as $Q1 X < Q3 Y$), segregation could not be made and the classifications of query cells in such cases were deemed confident. For example, all CD4+ T cells that are misclassified as CD8+ T cells will still obtain comparable IGS_{blood} with respect to correctly classified CD8+ T cells, both being subtypes of blood cells. If at any model throughout the hierarchy a low-confident IGS is observed, a flag is applied.”

4) For the benchmarking section, the authors should use other independent testing datasets instead of the training set of scATOMIC. Furthermore, the existing methods used for comparison are relatively old, and the authors should consider using some newly published methods to benchmark the performance of scATOMIC.

We would like to clarify that, indeed, Fig. 2a corresponds to an internal k-fold validation where testing was done on held out subsets of data from the same training source. However, in the other figures concerning the benchmarking process, Fig. 2b, 2c and Fig. 3, external datasets which are completely independent of those used to train scATOMIC were strictly used for testing. We now emphasize this in the figures’ descriptions and line 131 in the main text:

“We next aimed to conduct a comprehensive external, **training-independent** validation of scATOMIC performance.”

Regarding the reviewer’s second comment, as indicated in lines 145-149 we initially selected SingleR, SingleCellNet, scmap-cell, and CHETAH as they represent different approaches adapted by many other tools for cell type classification:

“These annotators encompass a variety of methods including reference correlation, label transfer using integration to a reference, random forest algorithm, flat and hierarchical models, and marker-based classification methods making their comparison with scATOMIC’s underlying RHC-REP approach highly suitable.”

The oldest tool, Scmap, was published in 2018 and together, the methods were cited close to 2K times. Due to the high interest of the community in using those tools and the relevant approaches they represent, we argue that they would be especially appropriate to include in the benchmarking section of the paper. Nonetheless, there are additional techniques that we failed to cover in the original version of the manuscript. We now include a comparison to the integration and label transfer approach provided by the Seurat package (Hao Y, *et al. Cell. 2021*, Stuart T, *et al. Cell. 2019*, >6K citations) and the most recently published marker gene-based method, scType (Ianevski A, *et al. Nature Communications. 2022*). In this analysis (updated Fig. 2c) we show comparable annotation accuracy among scATOMIC, Seurat, and scType in non-malignant cells. Nonetheless, as with the other tested methods, both the label transfer technique and the marker gene-based approach performed poorly in annotating malignant cells, especially compared to scATOMIC.

5) Line 173: the section titled “scATOMIC annotations increase cellular resolution in tumour biopsies” seems inappropriate. The classification achieved by scATOMIC is still relatively coarse-grained, while cell types such as pDC and mast cells may be also easily identified using traditional unsupervised clustering in conjunction with manually inspecting the expression of canonical marker genes. Furthermore, it is not clear whether scATOMIC still predict accurately if the model was trained for fine-grained cell types (like different T cell states).

We would like to thank the reviewer for this comment which encouraged us to evaluate additional extensions to scATOMIC’s core hierarchy. Due to the reasons mentioned in the main text, we aim to eliminate the need for manual inspection and provide accurate and automatic annotations for as many cell types as possible under the constraints of quality data availability (lines 36-42):

“Manual annotation is often unfavorable as it is subjective to user definition of non-parametric clustering of cells, conducted under the assumption that all the cells within a defined cluster are identical, and depends on pre-existing knowledge of canonical genes. Although expression of canonical markers has been used to characterize some cell types, definitive markers are not always available⁸. Moreover, due to their relatively low number and the possibility of incomplete detection due to technical variation, the sole use of canonical gene expression is not ideal.”

In addition to pDCs, in the updated version of scATOMIC, mast cells can now be accurately detected and annotated automatically. To date, we also added a new extension capable of sub-classifying T cells into CD4 naïve, CD4 effector/memory, CD4 Tfh/Th1, CD4 Treg, CD8 naïve, CD8 effector/memory, and CD8 exhausted T cells. We updated Fig. 1 and 4 with this expanded granularity, and we intend to update scATOMIC with additional models as soon as sufficient data for accurate model training and testing becomes available. In addition, we added a new figure (Extended Data Fig. 9) to highlight the overlap between the new T cell annotations and the expression of canonical marker genes.

Reviewer #2 (Remarks to the Author): Expert in single-cell RNA-seq, bioinformatics method development, and statistics

The authors present scATOMIC an automatic-annotation tool for single-cell RNAseq datasets derived from human cancer samples. The tool uses a novel hierarchical classification approach which they name "reversed hierarchical classification and repetitive elimination of parental nodes" which is based upon randomForest models trained at each node of a pre-determined cellular hierarchy. They train their method on a set of cancer cell-lines and then benchmark it against current state-of-the-art annotator on some flow-sorted data as well as compare it's ability to identify malignant cells to using CNV analysis. They demonstrate both that scATOMIC correctly annotates cells within its a priori defined hierarchy and that it will not incorrectly annotate cells with identities outside of the pre-defined hierarchy. Finally they present a use case for predicting the tumour-type of origin from metastatic cancer cells. Overall, it seems like an interesting and useful approach. I successfully installed the software and ran the tutorial.

We thank the reviewer for his comments and the time taken to review our manuscript. Responses are indicated in red, while references to the manuscript are shown in blue.

Major Concerns

1. The origin of the cellular hierarchy that is fundamental to the construction and operation of the method is unclear. Certain seemingly unrelated cancers are grouped together (e.g. kidney cancer grouped with Ovarian rather than Hepatobiliary and Pancreatic) , some leaves are specific cell-types while others are hybrid cell-type (e.g. Fibroblast/SmoothMuscleCell). Likewise distinct branches converge for CD8+ T cells which does not make sense give the logic of the approach. Could the authors clarify why they constructed the "tree" in this way and include in the manuscript guidance to users on how to construct their own trees for the application of expanding the method to additional cell-types.

We would like to thank the reviewer for this important comment. In summary, some linkages in the hierarchy are based on known relationships between cells (e.g., monocytes and macrophages) and cancer types (e.g., Colorectal, Esophageal and Gastric of the Gastrointestinal tract), while the transcriptomic similarity among other leaves in the final tree we agree was initially less apparent. Using a subset of our data corresponding to single patients, each with a different cancer, we trained a precursory version of the hierarchy to define strong transcriptomic connection where random forest models struggle (i.e., a similar percentage of the trees voting for different cancer). We are now clarifying how scATOMIC's hierarchy was generated in more detail in the Methods section (Defining the pan-cancer tumour microenvironment cell type hierarchy, lines 301-312):

"We defined a structured hierarchy where cell types with transcriptomic similarities are grouped into nodes (Fig. 1a). We first grouped blood cells based on existing relationships that correspond to the hematopoietic hierarchy, and kept stromal cells together. For cancer cells, we derived putative groups where transcriptomic similarities are expected based on the cancers' shared organ system, histological subtype, hormonal tissue, or germ layer. These included carcinomas of the digestive system (group 1: colorectal, gastric, esophageal, liver, gallbladder, bile duct, and pancreas), carcinomas not of the digestive system (group 2: lung, breast, prostate, endometrial, and ovarian), and non-carcinomas including soft tissue, neuroendocrine, and nervous system cancers (group 3 cancers: bone, sarcoma, brain, melanoma, and neuroblastoma). We then used a subset of our data, corresponding to one patient sample from each cancer type to generate random forest models. Evaluating the proportion of

trees voting for each cancer type helped refine the groups and provided guidelines concerning what the subsequent nodes might be.”

We specifically explained how this approach led us to include kidney in Group 2 tumours rather than in Group 1 with hepatobiliary and pancreatic cancers (lines 313-315):

“For example, for the less trivial grouping of kidney cancer, a classification model including all cancers assigned 42.3% of tree votes for kidney, while the majority of the remaining trees voted for various group 2 cancers thus, suggesting linking kidney cancer with group 2.”

We found at least one recent study supporting this linkage, reporting a lack of distinct transcriptome-based clustering between ovarian and kidney cancer of clear cell origin (Ji, J.X. *et al.*, The Journal of Pathology, 2018). Nonetheless, due to our technique of using a small data subset, we appreciate the possibility of other similar hierarchical structures.

With respect to the hybrid Fibroblast/Smooth Muscle leaf, this specific class has now been converted to Fibroblast only. By attempting to split the leaf into these two classes, it became apparent that the distinction among stromal cells in many published datasets is currently limited. This prevented the derivation of a reliable model trained for separating these cells and was the reason for our original decision to merge those classes. Along the same lines, retrospectively searching through our external validation datasets, we identified only one tumour that reported the presence of smooth muscle cells. Therefore, we decided to completely remove the smooth muscle notation. With the current version of scATOMIC, these more rare type of cells, in case interrogated, will be designated as fibroblasts. We intend to keep supporting scATOMIC with additional cell classes when sufficient data for accurate model training and testing becomes available.

Finally, concerning the last point in the reviewer’s comment, the hierarchy contains two converging cases, for lung cancer and CD8+ T cells, which are now discussed in the Methods section (lines 315-323):

“In the other case of lung cancer, we decided to include it in both groups 2 and 3, as separate cancers from both these groups obtained a high number of trees voting for lung. In this case, no clear distinction was observed. In a different case concerning CD8+ T cells, we included CD8+ T cells in both child nodes of the parent node T/NK as different CD8+ T cell populations show more transcriptomic similarities to NK (i.e. cytotoxic T cells) while others resemble CD4+ T cells more (Extended Data Fig. 1). We found that this structure yields stronger performance as compared to only having CD8+ cells in one of the branches or generating a single model to differentiate between CD4+, CD8+ and NK all at once.”

Supporting this, in the updated Extended Data Fig. 1, we show that effector and memory CD8+ T cells typically do not form a distinct cluster and fall in the space between CD4+ T and NK cells, while naïve CD8+ T cells cluster closer to CD4+ T cells. The CD8+ T cell location in the reduced multidimensional space can be partly explained by the expression of cytotoxic genes leading to overlaps with NK cells, while the expression of naïve T cell survival markers such as IL7R overlap with CD4+ T cells. Consequently, using two converging models helps to relax the complexity derived from cell classes representing multiple populations..

2. It is unclear how scATOMIC compares to SingleR, while the authors present SingleR as vastly inferior to scATOMIC in their benchmarking (Figure 2), but then show scATOMIC to perform nearly identically to SingleR in Figure 4.

To enable an unbiased comparison among tools, in Fig. 2, SingleR was trained on the same pan-cancer reference as scATOMIC. On the other hand, in Fig. 4a, the cells' labels were extracted directly from Slyper, M. *et al.*, 2020. In the latter case, the authors reported that they used SingleR, yet the reference they used was evidently much less inclusive than our own. Fig. 4 now further emphasizes the increased cellular resolution provided by scATOMIC with additional cell classes and an updated colour scheme. Of note, for others to benefit from our detailed pan cancer TME reference, we now share our fully annotated reference dataset (among other data used in the manuscript) through Zenodo (<https://doi.org/10.5281/zenodo.7419236>).

To mitigate potential misinterpretation of Fig. 4 purpose, we removed the word SingleR from the phrase "Original SingleR Annotations" in Fig. 4. In addition, we indicate the use of a different references in lines 183-185:

"Original annotations for this dataset were determined by the authors using SingleR¹⁵ with its default references in combination with cell type signatures and the use of canonical marker genes."

3. During the benchmarking the authors force scATOMIC to report terminal leaf annotations, however, in normal use the method would report internal nodes. What proportion of the time would the incorrect terminal annotations be reported by scATOMIC vs a correct though imprecise internal node? How often does scATOMIC report an internal node instead of the correct terminal node? Should users force scATOMIC to return terminal nodes for the best annotations?

We agree that the original text in the manuscript concerning the use of forced annotation was lacking. In the revised manuscript, the only time we forced terminal annotation is in Fig. 2c to enable a more fair comparison between scATOMIC and the other methods that are incapable of providing intermediate annotations. To address the reviewer's comment quantitatively, we analyzed the annotation of cells in the validation datasets, derived using either the unforced or forced mode. This analysis indicated that when unforced (default mode), scATOMIC assigned correct intermediate labels to 11% of incorrectly assigned cells in the forced mode. In comparison, by default scATOMIC only assigned intermediate labels to 2% of cells with correct terminal annotations obtained with the forced mode. These results argue in favour of using intermediate annotation. We now emphasize this at the beginning of the relevant Methods section "Comparison between scATOMIC and other tools" (lines 470-475):

"In this comparison only, we bypassed the use of IGS cut-offs in scATOMIC to enable comparison to other tools that cannot output intermediate cell classes. Of note, a separate analysis evaluating scATOMIC's performance under this forced mode against its default (unforced mode) favoured the latter by indicating that the use of intermediate annotations more frequently avoids derivation of incorrect terminal annotations than restricting the output of correct terminal classes."

4. Extended Data Figure 1 does not clearly support the conclusions for which it is cited in the manuscript. It shows several marker genes that can easily distinguish CD8+T, CD4+T, and NK cells as well as the fact that CD8+T uniquely express both genes expressed in CD4+T and genes expressed in NK cells. Yet the authors claim it shows that it is difficult to distinguish these cell-types based on marker genes.

We agree with the reviewer that our previous Extended Data Fig. 1 did not effectively convey the fact that in multi-cellular systems such as the tumour microenvironment, a stepwise classification and

elimination approach is needed. In the updated Extended Data Fig. 1, we now simply illustrate the idea of transcriptomic similarities among cells with CD8+ T cells as an example. When clustering CD8+ T cells, CD4+ T cells, and NK cells, we show that CD8+ T cells form two clusters corresponding to effector/memory CD8+ T cells and naïve CD8+ T cells. When we overlay the expression of IL7R, a gene that was a strong DEG for CD4+ T cells, we see that it has strong expression in both CD4 and two subpopulations of CD8 T cells. In contrast, the cytotoxic marker, GZMB, has overlapping expression among NK cells and a different CD8+ T cells subpopulation. This supports the derivation of multiple classification nodes, each representing more similar cells and having a more refined list of DEGs.

We note that this figure is cited in the text to illustrate some challenges involved in the derivation of cell-specific differentiating features which scATOMIC solves using stepwise classification. We modified lines 86-88 to link the updated Extended Data Fig. 1 to the main text better:

“Nonetheless, significant differentially expressed genes (DEG) concerning non-malignant cell types are often expressed in other related cells that are functionally distinct^{14,29,30} (Extended Data Fig. 1).”

5. Extended Data Figure 2 does not clearly support the conclusions for which it is cited in the manuscript. Panel c/d does seem to be somewhat clustered by cancer type - particularly the lower right corner (pink), lower left corner (yellowy-green) and top right corner (yellow-y orange). And panel e uses general biological processes vaguely related to cancer (some which are more related to the immune response to cancer not the cancer cells themselves - e.g. IFNresponse) not transcriptional programs derived from comparing malignant cancer cells to healthy control cells. Furthermore, despite claiming citing this figure that geneset signatures do not distinguish cancer cells and cancer cell-types they then use geneset signatures (from OncoDB) in the final stage of scATOMIC. If they used the geneset signatures from OncoDB in this figure would they see the same results?

We agree with the argument that transcriptomic heterogeneity in cell lines might be influenced by extended cell culture. Especially with respect to the reviewer’s following comment (#6), we have decided to remove panels c, d and e that correspond to cancer cell lines. In addition, we found these to be redundant considering the remaining panels, a and b, showing high and low transcriptomic heterogeneity in lung tumours among malignant and immune cells, respectively. In support of the high inter-patient heterogeneity observed in our data we also cite evidence of high transcriptomic heterogeneity in malignant cells as compared to non-malignant cells discussed with respect to at least three other cancers (melanoma: Tirosh, I. *et al. Science*, 2016, head and neck: Puram, S. V. *et al. Cell*, 2017, ovarian: Vázquez-García, I. *et al. Nature*, 2022). These references were included in our manuscript (lines 52, 87). We also modified lines 88-89 to link the updated Extended Data Fig. 2 to the main text better:

“On the other hand, inter-patient heterogeneity among malignant cells has been repeatedly observed with different patients forming unique clusters¹¹⁻¹³ (Extended Data Fig. 2).”

Regarding the reviewer’s inquiry about using OncoDB-derived signatures to cluster cancer cell lines, we want to clarify that while the biological processes from Kinker, *et al* (used in the original Extended Data Fig. 2e) represent a comparison between cancer types, OncoDB signatures were derived from cancer to matched normal tissue comparisons. It is therefore why scATOMIC uses OncoDB genes in the last step to differentiate cancer from normal. As many markers in OncoDB are core cancer genes shared among multiple cancer types, it is unlikely that OncoDB gene sets will provide the optimal solution to differentiate among cancers. To specifically address the reviewer’s comment, we reproduced the analysis as we did in the original Extended Data Fig. 2e, now using OncoDB genes. Scoring modules that are upregulated in cancers as compared to their matched normal was done

using the same method as was used in Kinker *et al.* for their reported recurrently heterogeneous programs.

We found that most cancers tend to form heterogeneous clusters (do not cluster by type), while other cancers (melanoma, breast, and colorectal cancer) show some level of grouping, but there were no fully homogeneous clusters by type (**Fig. R2**).

6. The authors claim: "inter-patient variability as a major obstacle for accurate classification" when considering their cell-line data. However, these cancer cell-lines have been grown for many generations in the lab, they are not primary cancer cells from patients. Thus they cannot conclude differences between the cell-lines result from inter-patient variability as genetic and transcriptional changes resulting from immortalization and in-vitro evolution are equally likely to cause the observed variability in expression profiles.

We agree with the reviewer. As mentioned in our response to the previous comment, we removed these figure panels and modified the text accordingly.

7. In the main text the authors claim: "RHC-REP prioritizes the features with the highest specificity to the interrogated cell types based on their absolute expression (Methods, Fig. 1b)" However, upon reading the Methods section I found that they identified features using differential expression testing of the SCTransformed gene expression values using the default wilcox test which in fact uses purely the relative rank of normalized expression values, not their absolute raw counts.

We initially used the words "based on their absolute expression" to indicate that instead of solely using fold change measurement to rank and prioritize genes as features, we used a second approach

(differential expression score, DES) to rank genes based on their specificity to any particular cell type. To avoid misinterpretation, the words "based on their absolute expression" were removed from the sentence in lines 101-102:

"For every model, we selected DEGs that distinguish each cell type from all other terminal classes nested within the same parent. RHC-REP will then prioritize the features with the highest specificity to the interrogated cell types (Fig. 1b)."

We also edited lines 346-349 in the Methods section ("Feature Selection") to reiterate the purpose of the DES step:

"We defined a differential expression score (DES) as the difference of the fraction of cells expressing a non-zero value for a respective DEG in ident.1 and ident.2 ($DES = pct_expr_ident.1 - pct_expr_ident.2$) to **capture DEGs more specifically expressed in any particular cell type.**"

Minor Concerns

8. The cell-type resolution of scATOMIC is quite low. In my experience identifying CD8+ T cells, CD4+ T cells and NK cells is simple to do and generally the type of cancer a dataset originates from is known a priori. The application to identifying malignant cells is more interesting and should be highlighted in the paper more.

To increase the cellular resolution output provided by scATOMIC we successfully trained additional accurate models to subclassify T cells into CD4 naïve, CD4 effector/memory, CD4 Tfh/Th1, CD4 Treg, CD8 naïve, CD8 effector/memory, and CD8 exhausted T cells and supported the new cell annotations using the expression of marker genes (Extended Data Fig. 9). We added emphasis in the discussion regarding the ability to identify malignant cells automatically aiding in downstream interpretation of data (lines 294-298):

"We expect that scATOMIC's abilities to accurately identify TME resident cells with high resolution, separate between cancer and normal tissue cells, and determine tumours' origin will enrich and expedite broad cancer studies seeking to refine prognostication or cell-cell communication from single-cell transcriptomes."

Additionally, in conjunction with our response to another reviewer's comment, we highlight another application of scATOMIC to identifying decreasing numbers of malignant cells (Extended Data Fig. 7).

9. In Figure 3b it is completely inappropriate to use a line graph. Line graphs imply relationships between adjacent points and in this case adjacent points derive from independent tumour samples.

We agree. The line graphs were replaced with barplots.

Reviewer #3 (Remarks to the Author): Expert in single-cell RNA-seq, bioinformatics, statistics, and tumour immune microenvironment

This manuscript introduces a new computational tool scATOMIC for classifying cells in tumor single-cell RNA-seq samples. Compared to existing tools, scATOMIC can more comprehensively annotate both malignant cells and non-malignant cells including immune and stromal cells. Through analysis of data from multiple tumor types, the authors demonstrated the ability of scATOMIC to better annotate cells in tumor samples with increased accuracy and resolution, and to infer tumor origins of metastatic samples. Overall, the manuscript is clearly written, and scATOMIC can be a useful tool for annotating cells in tumor single-cell RNA-seq samples. Below are several issues I suggest the authors to address to make the study more solid and useful.

We thank the reviewer for his comments and the time taken to review our manuscript. Responses are indicated in red, while references to the manuscript are shown in blue.

Major issues:

1. Data from different single-cell RNA-seq platforms and protocols have different characteristics. It is unclear whether the training and test data in this study are all from the same platform. If so, which platform? If not, the authors should provide the platform/protocol information for each dataset. In real applications, user's query data may be generated by a platform different from the one used to train the scATOMIC model. Can scATOMIC still produce reliable cell annotation in that scenario? How does the classification performance in query data vary across different platforms? A systematic benchmark to answer this question will be important.

We agree, please see our response to reviewer #1, comment #2 and the new Extended Data Fig. 5

2. The performance shown in this manuscript is on the cell-level. Does the cell classification accuracy vary a lot across samples? How much variation in performance should one expect across samples? Table S3 seems to suggest that the cell classification accuracy is very low in some samples. In real applications, how can users decide whether they should trust the predicted cell labels in their query sample (i.e. whether each query sample has high or low prediction accuracy)? Is there an objective statistic to guide this decision?

We agree that indications of low confident annotations might be beneficial to end users. To address this, we added a flag to scATOMIC final outputs that is based on defined thresholds derived by comparing the median IGS values at each classification node, for cells that were accurately annotated versus those that were misclassified during scATOMIC benchmarking. We provide details about this flag in the new Method section "Flagging cells with lower confidence annotations" (lines 407-424):

"Flagging cells with lower confidence annotations"

To provide a way for scATOMIC users to evaluate the confidence of their cell annotation output, we devised a secondary post-classification flag to warn about low-confidence annotations. To define low-confidence annotations, we used the results obtained by external validation (Fig. 2b). For every model throughout the hierarchy, we determined the median IGSs (or PS for terminal nodes) across samples for correctly annotated cells (X) and incorrectly annotated cells (Y). Correct versus incorrect status was defined by the terminal annotation of single cells. For example, in terminally annotated breast cancer cells, median $IGS_{non-blood}$, $IGS_{non-stromal}$, $IGS_{group2-cancer}$, $IGS_{breast/lung/prostate}$, PS_{breast} for all the correctly and incorrectly annotated cells in the validation data were recorded. We derived confidence thresholds based on the overlap between the distributions of X and Y using their quartiles (Q). When there was low overlap (defined as minimum $X > Q3 Y$), the threshold was set to the minimum X.

When there was intermediate overlap (defined as minimum $X < Q3 Y$, yet $Q1 X > Q3 Y$), the threshold was set to $Q3 Y$. In all remaining cases where there was high overlap (defined as $Q1 X < Q3 Y$), segregation could not be made and the classifications of query cells in such cases were deemed confident. For example, all $CD4+$ T cells that are misclassified as $CD8+$ T cells will still obtain comparable IGS_{blood} with respect to correctly classified $CD8+$ T cells, both being subtypes of blood cells. If at any model throughout the hierarchy a low-confident IGS is observed, a flag is applied.”

In addition, we would like to indicate that while outliers exist, our extensive training-independent validation analysis implies that users should expect low variation across samples (e.g., the maximum interquartile range of F1 scores shown in Fig. 2b is 0.034).

3. It is unclear what Extended Data Fig. 1 is showing. There is no description of the analysis procedure, making the data here hard to understand and reproduce.

We agree that the figure led to unnecessary confusion. The figure was intended to highlight that when considering all cells of the tumour microenvironment, the top DEGs for $CD8+$ T cells are also highly expressed in both $CD4+$ T and NK cells and by using stepwise classification, we mitigate the challenge of classifying them automatically, accurately. We recognize that the old figure was not ideal. In the updated Extended Data Fig. 1, we now simply illustrate the idea of transcriptomic similarities among cells with $CD8+$ T cells as an example. When clustering $CD8+$ T cells, $CD4+$ T cells, and NK cells, we show that $CD8+$ T cells form two clusters corresponding to effector/memory $CD8+$ T cells and naïve $CD8+$ T cells. When we overlay the expression of $IL7R$, a gene that was a strong DEG for $CD4+$ T cells, we see that it has strong expression in both $CD4$ and two subpopulations of $CD8$ T cells. In contrast, the cytotoxic marker, $GZMB$, has overlapping expression among NK cells and a different subpopulation of $CD8+$ T cells. This supports the derivation of multiple classification nodes, each representing more similar cells and having a more refined list of DEGs.

4. Figure 4 shows that scATOMIC can annotate cells with higher resolution compared to other annotations. A careful examination of this figure and Figure 1 suggests that the increased resolution is mainly for immune cells, and the resolution for cancer cell subtypes within each cancer type is still quite low. This seems to be a limitation of the current scATOMIC that should be discussed. It could be even better if the authors can systematically address this in a wide variety of cancer types like how they further classified breast cancer cell subtypes.

We would like to thank the reviewer for this comment which encouraged us to evaluate additional possible extensions to scATOMIC’s core hierarchy. While we agree that increased resolution is always preferable, there is currently not enough publicly available scRNA-seq data of specific cancer subtypes to derive accurate models, not the least to further test such new cancer subtypes extensions to reach the stringent standard we aim to provide in the other classification models we present. That being said, we are slightly confused by the reviewer considering this as a limitation. It is worth noting that there is currently no other annotation tool known to us, capable of identifying cancers’ tissue-of-origin at a greater resolution than that described in scATOMIC’s core hierarchy. Through the training and external validation of the extended breast cancer node, we demonstrate the possibility of adding additional cancer cell subtypes in the future. scATOMIC will be updated as more data becomes available. We now emphasize the above in the discussion section (lines 279-282):

“With the progressive accumulation of high quality publicly available scRNA-seq data, future extensions of the core hierarchy to further subclassify the other 18 cancer types and the various core non-malignant cell types to their more resolved classes or states will become simple.”

Moreover, we want to mention that as part of this revision process, we successfully increased the resolution of scATOMIC to include other non-malignant cell classes, where sufficient data were available (see updated Fig. 1a).

5. For the breast cancer analysis, the authors claim that "scATOMIC correctly subtyped 39 of the 40" biopsies. It is unclear how these statistics are obtained. In Table S5, there are 40 samples, and both of the HER2+/ER+ samples are incorrectly predicted as HER2+ instead of HER2+/ER+. This means the correctly subtyped biopsies have to be no more than 38. Where does the number 39 come from?

We thank the reviewer for pinpointing this error. We have changed the text to reflect that 38/40 samples were associated with classes that were represented in the scATOMIC breast cancer model and that 37/38 of those samples were correctly subtyped (lines 216-221):

We applied scATOMIC to the training-independent validation dataset containing 38 tumours spanning ER+, HER2+, and TN breast cancer, and 2 HER2+/ER+ double positive tumours, a class not represented in the current reference of scATOMIC's breast mode due to a lack of data. scATOMIC correctly subtyped 37 of the 38 (97.4%) training-independent breast cancer biopsies, as determined by immune-staining^{21,44} (Fig. 5b, Supplementary Table 5).

In addition, we clarify that the 2 Her2+ ER+ double positive samples that did not represent a class in the scATOMIC breast model received a mixture of HER2+ and ER+ annotations, suggesting a hybrid phenotype (lines 221-222):

"In the two HER2+/ER+ double positive samples, scATOMIC assigned mixed annotations of HER2+ and ER+ cells (Fig. 5b)."

6. It could be interesting to discuss or explore with data to show how the improved annotation of malignant and non-malignant cell types would allow one to better study interactions between different cell types (e.g. tumor cells and immune cells) in the TME.

As an example, following annotation with scATOMIC, we used CellChat (Jin, S., *et al. Nature Communications*, 2021) to interrogate lung and breast cancer biopsies in order to identify cell-cell interactions affecting cells at the gene pathway level (**Fig. R3**).

In lung cancer, we find significant interactions that are distinct to malignant cells as well as those involving normal tissue lung cells. Such a distinction between cancer and non-malignant cells that potentially share cell-of-origin is made possible through scATOMIC. The data suggests a strong interaction between lung cancer cells and normal tissue lung cells in the PTN pathway. This pathway was associated with tumour angiogenesis and cancer cell migration (Feng, ZJ., *et al. Oncogene*, 2010), suggesting that normal lung cells may also play a significant role in this behaviour of the tumour.

In another example involving breast cancer, we find significant interactions concerning the PDL1 pathways that were enriched between classical dendritic cells (cDCs) and exhausted CD8 T cells. Supporting this observation, subtypes of cDCs mediating T cell exhaustion through PD-L1-PD1 interactions have been recently reported (Oh, S.A., *et al. Nature Cancer* 2020). Such observations are made possible by the increased cell-type resolution of scATOMIC and can easily be obtained by leveraging its automatic classification capabilities.

We are glad that the reviewer raised this point. Indeed, we are planning to leverage scATOMIC for a systematic high-resolution characterization of cell-cell communication on a pan-cancer scale. This is a

large project that will necessitate extensive experimental validation of multiple communication channels that are beyond the scope of this work.

Starting the discussion, we argue that pan-cancer studies concerning tumour microenvironment interactions can benefit from suitable automatic annotation methods. A second statement suggesting leveraging scATOMIC in the context of cell-cell communication is now concluding the discussion (lines 294-298):

“We expect that scATOMIC’s abilities to accurately identify TME resident cells with high resolution, separate between cancer and normal tissue cells, and determine tumours’ origin will enrich and expedite broad cancer studies seeking to refine prognostication or cell–cell communication from single-cell transcriptomes.”

Figure. R3. Cell-cell communication pathway networks leveraging scATOMIC annotations.

Cell-cell communication networks were generated using CellChat for lung and breast cancers. Examples of significant pathways ($p < 0.05$) are shown. **a**, Probability of interactions between cell types in the PTN pathway in a lung cancer biopsy. The inferred source and target of interactions are shown. The strongest inferred interaction concerning the PTN pathway is between normal lung tissue cells and lung cancer cells. **b**, The inferred role of each cell type. **c**, Probability of interactions between cell types in the PDL1 pathway in a breast cancer. The inferred source and target of interactions are shown. CellChat infer immune checkpoint interactions between dendritic cells and exhausted CD8+ T cells. **d**, The inferred role of each cell type.

Reviewer #4 (Remarks to the Author): Expert in cancer genomics, bioinformatics, and machine learning

Nofech-Mozes et al. proposed a tool called scATOMIC for the annotation of malignant and non-malignant cells. The authors included a breast cancer dataset as a validation cohort with clinical relevance.

This is a highly relevant and well-conducted study and the paper is well-written. However, there is a lack of information on the manuscript that must be addressed before publication.

We thank the reviewer for his comments and the time taken to review our manuscript. Responses are indicated in red, while references to the manuscript are shown in blue.

Major:

1. The authors have to provide a schematic figure of the tool including the main function and the inputs-outputs.

We acknowledge the importance of adding this information. We addressed this in a new Supplementary Note 2. The note contains a schematic figure of scATOMIC. It describes the annotation process and contains descriptions of the main functions a user would call.

2. Did the authors check the performance of the tool before and after batch effect correction? In the manuscript, they mentioned the intra-patient similarities ("suggesting inter-patient variability as a major obstacle for accurate classification") so I am wondering if the performance of the tool can be improved using harmony with Patient ID as the cofactor.

scATOMIC transforms normalized read counts to relative expression values ranging from 0-1 and typically will select cell-type specific features that are ranked at the top. By ranking relative expression values during both training and implementation, we mitigate technical issues that occur across batches. In addition, we would like to refer the reviewer to our response to reviewer #1, comment #2 and the new Extended Data Fig. 5 addressing batch effects.

In a preliminary analysis using Harmony, we found that Harmony could not resolve cancer-type specific clusters (**Fig. R4**), suggesting no significant improvement in performance. Unfortunately, direct assessment of scATOMIC's performance following Harmony is not possible as Harmony corrects the PCA embeddings rather than outputting usable adjusted read counts. However, as part of this revision process, we evaluated scATOMIC performance against another approach (Seurat's rPCA integration pipeline followed by label transfer) that implements batch correction. Nonetheless, we found that it failed with cancer cells as compared to scATOMIC (Fig. 2c). Due to the extensive benchmarking conducted, we are highly confident in the performance of our tool.

Figure R4. Harmony correction does not resolve interpatient heterogeneity. Harmony integration was performed on cell lines with patient ID and technical batches as cofactors. UMAP of integrated data are coloured by **a**, cancer type, **b**, breast cancer cells, **c**, lung cancer cells, **d**, We confirmed that different inferred cell cycle phases did not confound clustering. Except for melanoma, cancers do not cluster by cancer type and are distributed in three distinct populations.

3. Regarding the comparison with other cell-type classification methods the authors have to present also the tool's computational time and the complexity of the proposed tool analysis for every module.

We added a new comparison illustrating the computational time and memory usage of all the tools tested (Extended Data Fig. 6). For all these comparisons, we used the same number of threads ($n=1$) with no restriction on memory usage. In this analysis, we found that scATOMIC performed faster than Seurat, SingleR, and CHETAH. We speculate that this is due to the fact that the other methods need to load the reference data and compare the query input rather than using a pre-trained model. SingleCellNet performed faster, likely due to the fact that it only uses a single model (flat model) while scATOMIC uses multiple models and Markov Affinity-based Graph Imputation of Cells which is a time-consuming step. scType performed fast as it only uses a set list of marker genes with no reference dataset or pre-trained model to function. We also compared the time and memory usage of

scATOMIC's module that differentiate cancer from normal tissue cells to CNV inference. While scATOMIC cancer signature is extremely fast compared to using CopyKAT to identify malignant cells, we note that CopyKat also provides the inferred CNVs, which is a larger layer of information.

4. The GitHub repository is very well structured and is ensured reproducibility. However, the data should be under a stable DOI (e.g. Zenodo). Moreover, a short description of the main functions is necessary to be described also to the manuscript (e.g. as supplementary material)

To support reproducibility further, we followed this reviewer's suggestion and added the entire package, training data, validation data and the code to generate the main figures to Zenodo (<https://doi.org/10.5281/zenodo.7419236>). In addition, a description of the main functions can be found in the new Supplementary Note 2.

5. The study will gain a lot of power in their approach if the authors include as an additional clinical scenario the measurable residual disease after treatment.

We conducted a new analysis that reflects scATOMIC's sensitivity to detect cancer cells by simulating residual disease (lines 170-173):

"in silico serial dilution analysis of cancer cells in our external dataset collection revealed that decreasing number of malignant cells can be appropriately annotated, with scATOMIC also providing cancer type notations (Extended Data Fig. 7)."

In the new Extended Data Fig. 7, we show that scATOMIC can consistently identify as low as 10 cancer cells among non-malignant cells across heterogenous TME. We appreciate that the future evolution of scATOMIC from a research tool to a diagnostic one will require additional work and substantial amounts of clinical-grade data to support clinical scenarios such as the one raised by the reviewer.

Minor:

1. line 42: has to have

We disagree with this grammar suggestion as the "has" is referring to the term expression which is in singular form.

2. line 93: remove As

This sentence has been changed.

3. The link to the GitHub repo does not work

We are unclear why it did not work. It may have been an issue with the document conversion in the submission process. We tested it again, made sure that the repository is public and validated access by external users.

4. In figure 5, the authors have to include the number of cells per patient and the predicted number of cells in each case

We added the overall number of cells corresponding to each patient dataset and the numbers of cells that were annotated by scATOMIC as breast cancer cells to the related Supplementary Table 5.

REVIEWER COMMENTS

Reviewer #1 (Remarks to the Author):

The authors have addressed all of my comments and concerns.

Reviewer #2 (Remarks to the Author):

I thank the authors for their clarifications the manuscript is much improved.

Reviewer #3 (Remarks to the Author):

The authors have addressed most of my questions. My question #2 was not fully addressed though. I was asking about how prediction accuracy varies across samples (i.e. interpatient variability of prediction accuracy). For example, for one patient, most cells may be classified accurately. For another patient, most cells may be classified incorrectly. Then in a hypothetical future clinical/precision healthcare setting, one may have to recollect and reanalyze the sample from the second patient. How big is such variability? When there is a new patient sample, is there a metric that tells users how confident they are about the predictions on that sample? The authors response only discussed confidence at the cell level, but not at the sample (i.e. patient) level. The latter is what I was asking about.

Reviewer #4 (Remarks to the Author):

Thank you for addressing my points.

Reviewer #3 (Remarks to the Author):

The authors have addressed most of my questions. My question #2 was not fully addressed though.

We appreciate the clarification. Our response consists of two parts. The first concerns the reviewer's query about inter-patient variability in our data and the second concerns an output aimed at providing users with sample-level confidence. Responses are indicated in red, while references to the manuscript are shown in blue.

I was asking about how prediction accuracy varies across samples (i.e. interpatient variability of prediction accuracy). For example, for one patient, most cells may be classified accurately. For another patient, most cells may be classified incorrectly. Then in a hypothetical future clinical/precision healthcare setting, one may have to recollect and reanalyze the sample from the second patient. How big is such variability?

To highlight the variability in performance of scATOMIC at a sample level we calculated weighted F1 scores corresponding to the mean F1 scores for each cell type in a respective sample, weighted by the number of cells. As shown in the newly generated **Extended Data Fig. 12**, there is little performance variability across samples (Interquartile range of 0.045, ranging from quartile1 = 0.946 and quartile3 = 0.991).

When there is a new patient sample, is there a metric that tells users how confident they are about the predictions on that sample? The authors response only discussed confidence at the cell level, but not at the sample (i.e. patient) level. The latter is what I was asking about.

Since prediction accuracy calculation relies on ground-truth cell identities, it will be impossible to directly measure it in future samples. Nonetheless, we can provide estimates of sample level F1 values. We previously detailed a confidence parameter at the level of cells (**methods section titled "Flagging cells with lower confidence annotations"**). We now extended this metric to the sample level, to summarize the proportion of cells receiving a confident annotation in any given sample. There is a significant positive correlation between predicted sample-level confidence values and the weighted F1 scores in the samples used for scATOMIC external validation (n=226, Pearson's R=0.69, P-value<2.2e-16). Nonetheless, since there is little variability across the samples in their weighted F1 scores, and correlation is driven mostly by high values, we decided on using a cut-off for reporting low-confident, sample level flags (**Extended Data Fig. 12**). The cut-off was set to prioritize sensitivity in detecting problematic samples. As shown by the dashed line in the new figure, 11 out of the 226 samples in our external validation datasets (<5%) had a weighted F1 lower than 0.75 indicating lower prediction accuracy. 100% of those 11 samples also had less than 75% of their cells received confidence annotations based on our previously reported flag (i.e., 100% sensitivity). In contrast, only 5.58% (12/215) of samples with per-sample confidence values < 0.75 had weighted F1 score > 0.75.

This new flag is expected to capture all problematic samples to encourage user to re-evaluate the integrity of their generated raw scRNA-seq data.

We added the following to the end of the **methods section titled "Flagging cells with lower confidence annotations"**: In addition, we assigned a sample-level confidence metric derived from the proportion of cells receiving a confident annotation based on this flag. To maximize identification of potentially poor-quality samples, in any new interrogated sample, if less than 75% of cells receive confident annotations, a warning will be issued (**Extended Data Fig. 12**).

Extended Data Fig. 12. Inter-sample variability in scATOMIC performance. Weighted F1 scores are plotted for each sample in the external validation dataset (n=226). Each sample (dot) is coloured by its assigned classification confidence status derived from the proportion of confidently annotated cells. All of the samples with a weighted F1 score < 0.75 also had less than 75% of cells obtaining high confident annotations. Boxes and whiskers represent the lower fence, first quartile (Q1), median (Q2), third quartile (Q3), and upper fence.

REVIEWERS' COMMENTS

Reviewer #3 (Remarks to the Author):

The authors have satisfactorily addressed my remaining questions.